# Determinantal Point Process Selection Over Grid Codes Supports Out of Distribution Generalization

## Abstract

Deep neural networks have made tremendous gains in achieving human-like intelligence, but still lag behind human competence in strong forms of generalization. One such case is out-of-distribution (OOD) generalization— successful performance on test examples that lie outside the distribution of the training set. Here, we identify how certain properties of processing in the brain can be used to achieve strong OOD generalization and offer a two-part algorithm that improves OOD generalization performance of artificial neural networks on multiple tasks widely used in neuroscience to demonstrate proof of concept. First, we exploit the fact that the mammalian brain represents metric spaces using grid-like representations: abstract representations of relational structure, organized in recurring motifs that cover the representational space. Second, we propose a selectional mechanism that operates over these grid representations using determinantal point process (DPP-S) - a transformation that ensures maximum sparseness in the coverage of that space. We show that a loss function that combines standard task-optimized error with DPP-S can exploit the recurring motifs in grid codes, and can be integrated with common architectures to achieve strong OOD generalization performance on analogy and arithmetic tasks.

## 1 Introduction

Deep neural networks now meet, or even exceed, human competency in many challenging task domains (He et al., 2016; Silver et al., 2017; Wu et al., 2016; He et al., 2017). Their success on these tasks, however, is generally limited to the narrow set of conditions under which they were trained, falling short of the capacity for out-of-distribution (OOD) generalization that is central to human intelligence (Barrett et al., 2018; Lake & Baroni, 2018; Hill et al., 2019; Webb et al., 2020). Here, we consider two cognitive problems that are widely used in neuroscience and often require a capacity for OOD generalization: a) analogy and b) arithmetic. What enables the human brain to successfully generalize on these tasks, and how might we better realize that ability in deep learning systems?

To address the problem, we focus on two properties that we hypothesize are useful for OOD generalization in biological systems: a) the *representations* of relational structure, in which relations are preserved across transformations like translation and scaling (such as observed for grid cells in mammalian medial entorhinal cortex (Hafting et al., 2005)); and b) a *selectional objective* to select representations that have maximum variance and minimum correlation among them, over the training data. The net effect of these two properties is to normalize the representations of training and testing data in a way that preserves their relational structure, and allows the network to learn that structure in a form that can be applied well beyond the domain over which it was trained.

In previous work, it has been shown that such OOD generalization can be accomplished in a neural network by providing it with a mechanism for temporal context normalization (Webb et al., 2020), a technique that allows neural networks to preserve the relational structure between the inputs in a local temporal context, while abstracting over the differences between contexts. Here, we test whether the same capabilities can be achieved using a well-established, biologically plausible embedding scheme – grid codes – and an adaptive form of normalization that is based strictly on the statistics of the training data in the embedding

space. We show that when deep neural networks are presented with data that exhibits such relational structure, grid code embeddings coupled with an error-minimizing/selectional objective promotes strong OOD generalization. We unpack each of these theoretical components in turn before describing the tasks, modeling architectures, and results.

**Representations of Relational Structure.** The first component of the proposed framework relies on the idea that a key element underlying human-like OOD generalization is the use of low-dimensional representations that emphasize the relational structure between data points. Empirical evidence suggests that, for spatial information, this is accomplished in the brain by encoding the organism's spatial position using a periodic code consisting of different frequencies and phases (akin to a Fourier transform of the space). Although grid cells were discovered for representations of space (Hafting et al., 2005), they have since been identified in non-spatial domains, such as auditory tones (Aronov et al., 2017), odor (Bao et al., 2019), and conceptual dimensions (Constantinescu et al., 2016). These findings suggest that the coding scheme used by grid cells may serve as a general representation of metric structure that may be exploited for reasoning about the abstract conceptual dimensions required for higher level reasoning tasks, such as analogy and mathematics (McNamee et al., 2022). Of interest here, the periodic response function displayed by grid cells belonging to a particular frequency is invariant to translation by its period, and increasing the scale of a higher frequency response gives a lower frequency response and vice versa, making it invariant to scale across frequencies. This is particularly promising for prospects of OOD generalization: downstream systems that acquire parameters over a narrow training region may be able to successfully apply those parameters across transformations of translation or scale, given the shared structure (which can also be learned (Cueva & Wei, 2018; Banino et al., 2018; Whittington et al., 2020)).

**DPP Selection (DPP-S).** The second component of our proposed framework is a novel selectional objective that uses the statistics of the training data to sculpt the influence of grid cells on downstream computation. Despite the use of a relational encoding metric (i.e., grid code), generalization may also require identifying which aspects of this encoding that could potentially be shared across training and test distributions. Here, we implement this by identifying, and restricting further processing to those grid embeddings that exhibit the greatest variance, but are least redundant (that is, pairwise uncorrelated) over the training data. Formally, this is captured by maximizing the determinant of the covariance matrix of the grid embeddings computed over the training data (Kulesza & Taskar, 2012). To avoid overfitting the training data, we select a subset of grid embeddings that maximize the volume in the representational space, diminishing the influence of low-variance codes (irrelevant), or codes with high-similarity to other codes (redundant), which decrease the determinant of the covariance matrix. We refer to this as DPP selection, or DPP-S.

DPP-S is inspired by mathematical work in statistical physics using Determinantal Point Processes (DPPs) that originated for modeling the distribution of fermions at thermal equilibrium (Macchi, 1975). DPPs have since been adopted in machine learning for applications in which diversity in a subset of selected items is desirable, such as recommender systems (Kulesza & Taskar, 2012). Recent work in computational cognitive science has shown DPPs naturally capture inductive biases in human inference, such as some word-learning and reasoning tasks (e.g., one noun should only refer to one object) while also serving as an efficient memory code (Frankland & Cohen, 2020). In that context, the learner is biased to find a set of possible word-meaning pairs whose representations exhibit the greatest variance and lowest covariance on a task-relevant dataset. DPPs also provide a formal objective for the type of orthogonal coding that has been proposed to be characteristic of representations in mammalian hippocampus, and integral for episodic memory (McClelland et al., 1995). Thus, using the DPP objective to govern selection of grid code representations, known to be implemented in the entorhinal cortex (one synapse upstream of the hippocampus), aligns with the function and organization of cognitive and neural systems underlying the capability for abstraction.

Taken together, the representational and selection mechanisms outlined above define a two-component framework for promoting OOD generalization, by minimizing task-specific error subject to: i) embeddings that encode relational structure among the data (grid cells), and ii) selection of those embeddings that maximize the "volume" of the representational space that is covered, while minimizing redundancy (DPP-S). Below, we demonstrate proof of concept by showing that these mechanisms allow artificial neural networks to learn representations that support OOD generalization on two cognitive tasks which have similarly been the focus

in neuroscience and therefore serve as a reasonable starting point for examining the properties of interest in these networks.

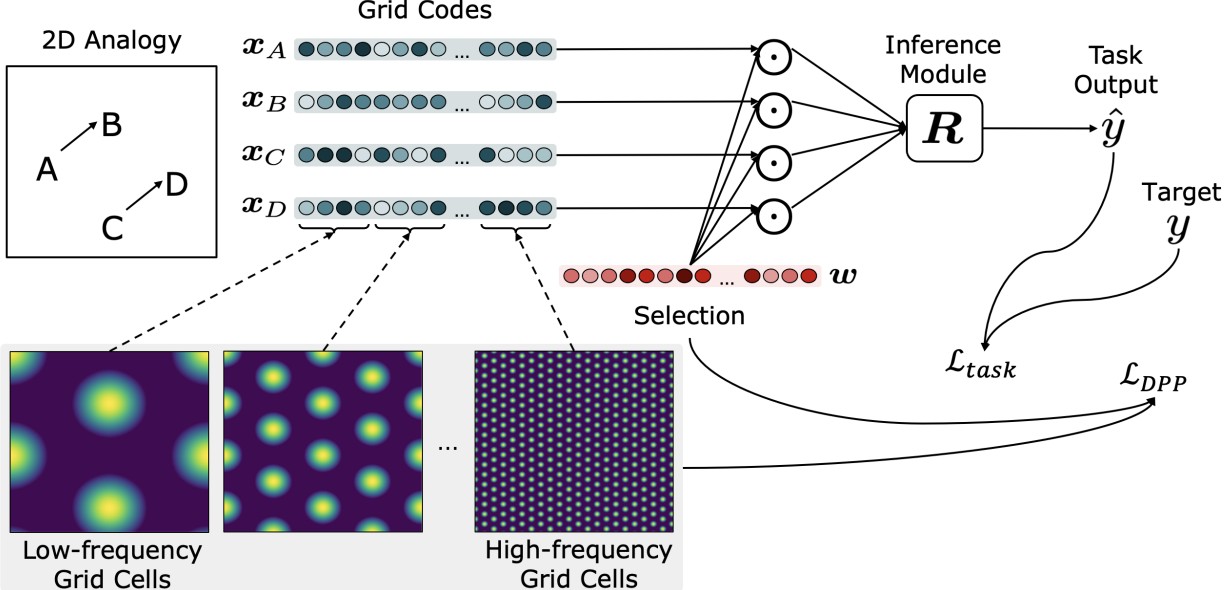

Figure 1: Schematic of the overall framework. Given a task (e.g., an analogy to solve), inputs (denoted as $\{A, B, C, D\}$) are represented by grid codes, consisting of units ("grid cells") representing different combinations of frequencies and phases. Grid embeddings $(\boldsymbol{x}_A, \boldsymbol{x}_B, \boldsymbol{x}_C, \boldsymbol{x}_D)$ are multiplied elementwise by a set of learned selection weights $\boldsymbol{w}$, then passed to a inference module $\boldsymbol{R}$. The selection weights $\boldsymbol{w}$ are optimized using $\mathcal{L}_{DPP}$, which encourages selection of grid embeddings that maximize the volume of the representational space. The inference module outputs a score for each candidate analogy (consisting of $A, B, C$ and a candidate answer choice $D$). The scores for all answer choices are passed through a softmax to generate an answer $\hat{y}$, which is compared against the target $y$ to generate the task loss $\mathcal{L}_{task}$.

## 2 Approach

Figure 1 illustrates the general framework. Task inputs, corresponding to points in a metric space, are represented as a set of grid code embeddings $\boldsymbol{x}_{t=1..T}$, that are then passed to a inference module $\boldsymbol{R}$. The embedding of each input is represented by the pattern of activity of grid cells that respond selectively to different combinations of phases and frequencies. Selection over these is a learned weighting $\boldsymbol{w}$ of the grid cells, the weighted activations of which $(\boldsymbol{x} \odot \boldsymbol{w})$ are passed to the inference module $(\boldsymbol{R})$. The parameterization of $\boldsymbol{w}$ and $\boldsymbol{R}$ are determined by backpropagation of the error signal obtained by two loss functions over the training set. The first, $\mathcal{L}_{DPP}$ favors selectional weightings over the grid cells that maximize the DPP-S objective; that is, the "volume" of the representational space (grid code) covered by the selected grid cells. The second, $\mathcal{L}_{task}$ is a standard task error term (e.g., the cross entropy of targets $y$ and task outputs $\hat{y}$ over the training set). We describe each of these components in the following sections.

### 2.1 Task setup

#### 2.1.1 Analogy task

We constructed proportional analogy problems with four terms, of the form $A : B :: C : D$, where the relation between $A$ and $B$ was the same as between $C$ and $D$. Each of $A, B, C, D$ was a point in the integer space $\mathbb{Z}^2$, with each dimension sampled from the range $[0, M-1]$, where $M$ denotes the size of the training region. To form an analogy, two pairs of points $(A, B)$ and $(C, D)$ were chosen such that the vectors $AB$ and $CD$ were equal. Each analogy problem also contained a set of 6 foil items sampled in the range $[0, M-1]^2$

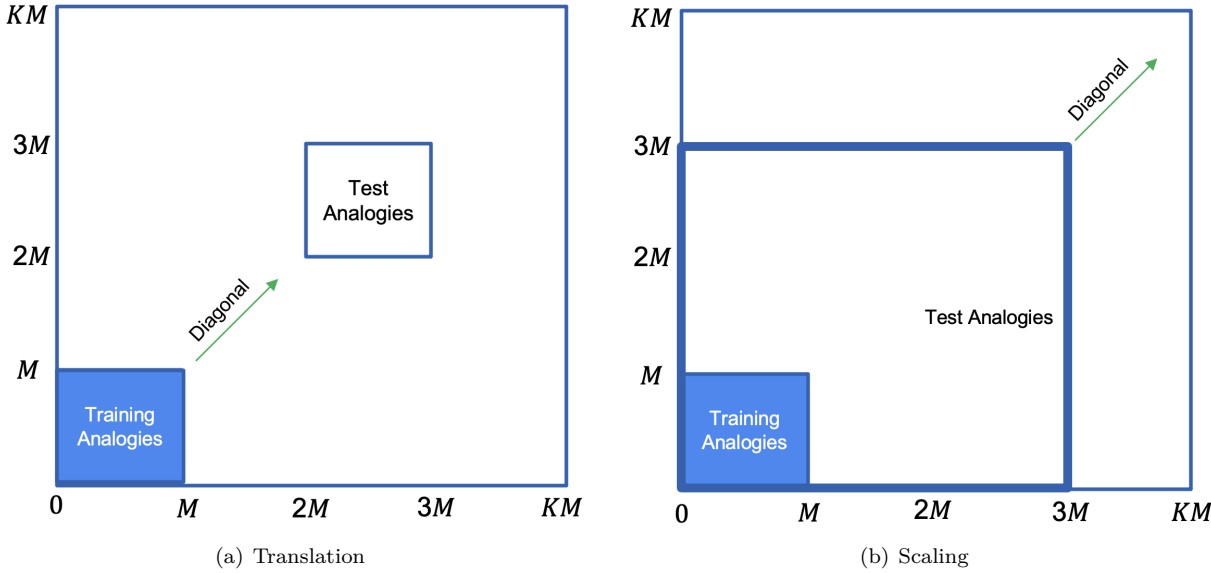

Figure 2: Generation of test analogies from training analogies (region marked in blue) by: a) translating both dimension values of $A, B, C, D$ by the same amount; and b) scaling both dimension values of $A, B, C, D$ by the same amount. Since both dimension values are transformed by the same amount, each input gets transformed along the diagonal.

excluding $D$, such that they didn't form an analogy with $A, B, C$. The task was, given $A$, $B$ and $C$, to select $D$ from a set of multiple choices consisting of $D$ and the 6 foil items. During training, the networks were exposed to sets of points sampled uniformly over locations in the training range, and with pairs of points forming vectors of varying length. The network was trained on 80% of all such sets of points in the training range, with 20% held out as the validation set.

To study OOD generalization, we created two cases of test data, that tested for OOD generalization in translation and scale. For the *translation invariance* case (Figure 2(a)), the constituents of the training analogies were translated along both dimensions by the same amount [1] $KM$ such that the test analogies were in the range $[KM, (K+1)M - 1]^2$ after translation. Non-overlapping test regions were generated for $K \in [1, 9]$. Similar to the translation OOD generalization regime of Webb et al. (2020), this allowed the graded evaluation of OOD generalization to a series of increasingly remote test domains as the distance from the training region increased. For example a training analogy $A : B :: C : D$ after translation by $KM$, would be $A + KM : B + KM :: C + KM : D + KM$. For the *scale invariance* case (Figure 2(b)), we scaled each constituent of the training analogies by $K$ so that the test analogies after scaling were in the range $[0, KM - 1]^2$. Thus, an analogy $A : B :: C : D$ after scaling by $K$, would be $KA : KB :: KC : KD$. By varying the value of $K$ from 1 to 9, we scaled the training analogies to occupy increasingly distant and larger regions of the test space.

### 2.1.2 Arithmetic task

We tested two types of arithmetic operations, corresponding to the translation and scaling transformations used in the analogy tasks: elementwise addition and multiplication of two inputs $A$ and $B$, each a point in $\mathbb{Z}^2$, for which $C$ was the point corresponding to the answer (i.e., $C = A + B$ or $C = A * B$). As with the analogy task, each arithmetic problem also contained a set of 6 foil items sampled in the range $[0, M-1]^2$, excluding $C$. The task was to select $C$ from a set of choices consisting of $C$ and the 6 foil items. Similar to the analogy task, training data was constructed from a uniform distribution of points and vector lengths in the training range,

---

[1]We transformed by the same amount along both dimensions so that the OOD generalization regimes are similar to Webb et al. (2020).

with 20% held out as the validation set. To study OOD generalization, we created testing data corresponding to $K = 9$ non-overlapping regions, such that $C \in [M, 2M - 1]^2, [2M, 3M - 1]^2, ...[KM, (K + 1)M - 1]^2$.

## 2.2 Architecture

### 2.2.1 Grid codes

As discussed above, grid codes are found in the mammalian neocortex, that support structured, low-dimensional representations of task-relevant information. For example, an organism's location in 2D allocentric space (Hafting et al., 2005), the frequency of 1D auditory stimuli (Aronov et al., 2017), and conceptual knowledge in two continuous dimensions (Constantinescu et al., 2016) have all been shown to be represented as unique, similarity-preserving combinations of frequencies and phases. Here, these codes are of interest because the relational structure in the input is preserved in the code across translation and scale. This offers a promising metric that can be used to learn structure relevant to the processing of analogies (Frankland et al., 2019) and arithmetic over a restricted range of stimulus values, and then used to generalize such processing to stimuli outside of the domain of task training.

To derive grid codes for stimuli, we follow the analytic approach described by Bicanski & Burgess (2019) [2]. Specifically, the grid code ($\boldsymbol{x}$) for a particular stimulus location $A$ is given by:

$$\boldsymbol{x}_A = max(0, cos(\boldsymbol{z}_0) + cos(\boldsymbol{z}_1) + cos(\boldsymbol{z}_2)) \tag{1}$$

where,

$$\boldsymbol{z}_i = \boldsymbol{b}_i * (FA + A_{offset}) \tag{2}$$

$$\boldsymbol{b}_0 = \begin{pmatrix} cos(0) \\ sin(0) \end{pmatrix}, \boldsymbol{b}_1 = \begin{pmatrix} cos(\frac{\pi}{3}) \\ sin(\frac{\pi}{3}) \end{pmatrix}, \boldsymbol{b}_2 = \begin{pmatrix} cos(\frac{2\pi}{3}) \\ sin(\frac{2\pi}{3}) \end{pmatrix} \tag{3}$$

The spatial frequencies of grids ($F$) begin at a value of $0.0028 * 2\pi$. Wei et al. (2015) have shown that, to minimize the number of variables needed to represent an integer domain of size $S$, the firing rate widths and frequencies should scale geometrically in a range ($\sqrt{2}, \sqrt{e}$), closely matching empirically observed scaling in entorhinal cortex (Stensola et al., 2012). We choose a scaling factor of $\sqrt{2}$ to efficiently tile the space. One consequence of this efficiency is that the total number of discrete frequencies in entorhinal cortex is expected to be small. Empirically, it has been estimated to be between 8-12 (Moser et al., 2015) [3]. Here, we choose $N_f = 9$ (dimension of $F$) as the number of frequencies. $A$ refers to a particular location in a two dimensional space, and 100 offsets ($A_{offset}$) are used for each frequency to evenly cover a space of $1000 \times 1000$ locations using 900 grid cells. These offsets represent different phases within each frequency and since there are 100 of them, $N_p = 100$. Each point from the set of 2D points for the stimuli in a task (described in Section 2.1), was represented using the firing rate of 900 grid cells which constituted the grid embedding for that point to form the inputs to our model.

### 2.2.2 DPP-S

We hypothesize that the use of a relational encoding metric (i.e., grid code) is extremely useful, but not sufficient for a system to achieve strong generalization, which requires selecting particular aspects of the encoding that can capture the same relational structure across the training and test distributions. Toward this end, we propose a selectional objective that uses the statistics of the training data to select the grid embeddings that can induce the inference module to achieve strong generalization. Our objective, which we describe in detail below, seeks to identify those grid embeddings that exhibit the greatest variance but are least redundant (pairwise uncorrelated over the training data). Formally, this is captured by maximizing the determinant of the covariance matrix of the grid embeddings computed over the training data (Kulesza

---

[2]https://github.com/bicanski/VisualGridsRecognitionMem

[3]It seems likely that the use of grid codes for abstraction in human cognition requires a considerably greater number of states $S$ than that used by the rodent for sensory encoding. However, given exponential scaling, the total number of frequencies is expected to remain low, increasing as a logarithm of $S$.

& Taskar, 2012). Although in machine learning, DPPs have been particularly influential in work on recommender systems (Chen et al., 2018), summarization (Gong et al., 2014; Perez-Beltrachini & Lapata, 2021), neural network pruning (Mariet & Sra, 2015), here, we propose to use maximization of the determinant of the covariance matrix as a filtering mechanism to limit the influence of grid embeddings with low-variance (which are less relevant) or with high-similarity to other grid embeddings (which are redundant).

For the specific tasks that we study here, we have assumed the grid embeddings to be pre-learned to represent the entire space of possible test data points, and we are simply focused on the problem of how to determine which of these are most useful in enabling generalization for a task-optimized network trained on a small fraction of that space (Figure 2). That is, we look for a way to select a subset of grid-cells frequencies whose embeddings capture recurring task-relevant relational structure. We find that grid embeddings corresponding to the higher spatial frequency grid cells exhibit more variance on the training data than the low frequency embeddings. In any training set, low spatial frequency embeddings may carry information about the stimuli that can be used to help minimize task-error, but, critically, the higher frequency embeddings due to their higher variance tend to preferentially capture the same relational structure across different regions which is necessary for OOD generalization. Accordingly, we find that filtering for the determinant maximizing grid cell embeddings tends to select those that encode higher frequencies.

Formally, we treat obtaining $\mathcal{L}_{DPP}$ as an approximation of a determinantal point process (DPP). A DPP $\mathcal{P}$ defines a probability measure on all subsets of a set of items $\mathcal{X} = \{1, 2, \ldots N\}$. For every $\boldsymbol{x} \subseteq \mathcal{X}$, $P(\boldsymbol{x}) \propto \det(\boldsymbol{V_x})$. Here $\boldsymbol{V}$ is a positive semidefinite covariance matrix and $\boldsymbol{V_x} = [V_{ij}]_{i,j \in \boldsymbol{x}}$ denotes the matrix $\boldsymbol{V}$ restricted to the entries indexed by elements of $\boldsymbol{x}$. The maximum a posteriori (MAP) problem $argmax_{\boldsymbol{x}} \det(\boldsymbol{V_x})$ is NP-hard (Ko et al., 1995). However $f(\boldsymbol{x}) = \log(\det(\boldsymbol{V_x}))$ satisfies the property of a submodular function, and various algorithms exist for approximately maximizing them. One common way is to approximate this discrete optimization problem by replacing the discrete variables with continuous variables and extend the objective function to the continuous domain. Gillenwater et al. (2012) proposed a continuous extension that is efficiently computable and differentiable:

$$\hat{F}(\boldsymbol{w}) = \log \sum_{\boldsymbol{x}} \prod_{i \in \boldsymbol{x}} w_i \prod_{i \notin \boldsymbol{x}} (1 - w_i) \exp(f(\boldsymbol{x})), \boldsymbol{w} \in [0, 1]^N. \tag{4}$$

We use the following theorem from Gillenwater et al. (2012) to construct $\mathcal{L}_{DPP}$:

**Theorem 2.1.** *For a positive semidefinite matrix $\boldsymbol{V}$ and $\boldsymbol{w} \in [0, 1]^N$:*

$$\sum_{\boldsymbol{x}} \prod_{i \in \boldsymbol{x}} w_i \prod_{i \notin \boldsymbol{x}} (1 - w_i) \det(\boldsymbol{V_x}) = \det(\text{diag}(\boldsymbol{w})(\boldsymbol{V} - \boldsymbol{I}) + \boldsymbol{I}) \tag{5}$$

We propose a selection mechanism that uses $\mathcal{L}_{DPP}$ to select subsets of grid embeddings for further processing. Algorithm 1 describes the training procedure with DPP-S which consists of two steps, using $\mathcal{L}_{DPP}$ as the first step. This step maximizes the objective function:

$$\hat{F}(\boldsymbol{w}, \boldsymbol{V}, N_f, N_p) = \sum_{f=1}^{N_f} \log \det(\text{diag}(\boldsymbol{w_f})(\boldsymbol{V_f} - \boldsymbol{I}) + \boldsymbol{I}) \tag{6}$$

using stochastic gradient ascent for $N_{E_{DPP}}$ epochs, which is equivalent to minimizing $\mathcal{L}_{DPP}$, as $\mathcal{L}_{DPP} = -\hat{F}(\boldsymbol{w}, \boldsymbol{V}, N_f, N_p)$. It involves selecting the grid embeddings that exhibit the greatest within frequency variance but are least redundant (that is, that are least also pairwise uncorrelated) over the training data. This is achieved by maximizing the determinant of the covariance matrix over the within frequency grid embeddings of the training data, which is obtained by applying log on both sides of the Theorem 2.1, and in our case $\boldsymbol{x}$ refers to grid cells within a particular frequency. Here $\boldsymbol{w}$ are weights corresponding to each grid cell, and $N_f$ is the number of distinct frequencies. The matrix $\boldsymbol{V}$ captured a measure of the covariance of the grid embeddings over the training region. We used the *synth_kernel* function [4] to construct $\boldsymbol{V}$,

---

[4] https://github.com/insuhan/fastdppmap/blob/db7a28c38ce654bdbfd5ab1128d3d5910b68df6b/test_greedy.m#L123.
$S$ need not be a square matrix in our case, whose second dimension $M$ was the size of the training region. *L_kernel* is same as $\boldsymbol{V}$.

where in our case $\boldsymbol{m}$ are the variances of the grid cell representations $\boldsymbol{S}$ computed over the training region $M$, $N$ is the number of grid cells and $w_m, b$ are hyperparameters with values of 1 and 0.1 respectively. The dimensionality of $\boldsymbol{V}$ was $N_f N_p \times N_f N_p (900 \times 900)$. $\boldsymbol{w_f}$ were the weights of the grid cells belonging to the $f$th frequency , so $\boldsymbol{w_f} = \boldsymbol{w}[fN_p : (f+1)N_p]$, where $N_p$ was the number of phases for each frequency. $\boldsymbol{V_f} = \boldsymbol{V}[fN_p : (f+1)N_p, fN_p : (f+1)N_p]$ was the restriction of $\boldsymbol{V}$ to the grid embeddings for $f$th frequency, so it captured the covariance of the grid embeddings belonging to the $f$th frequency. Equation 6 which involved summation of the logarithm of the determinant of the weighted covariance matrix of grid cells within each frequency, over $N_f$ frequencies was used to compute the negative of $\mathcal{L}_{DPP}$. Maximizing $\hat{F}$ gave the approximate maximum within frequency log determinant for each frequency $f \in [1, N_f]$, which we denote for the $f$th frequency as $\hat{F}_f$. In the second step of the training procedure, we used the $f_{max_{DPP}}$ grid cell frequency, where $f_{max_{DPP}} = \arg\max_{f \in [1, N_f]} \hat{F}_f$. In other words, we used the grid embeddings for grid cells belonging to the frequency which had the maximum within-frequency log determinant at the end of the first step, which we find are best at capturing the relational structure across the training and testing data, thereby promoting out-of-distribution generalization. In this step, we trained the inference module minimizing $\mathcal{L}_{task}$ over $N_{E_{task}}$ epochs.

### 2.2.3 Inference module

We implemented the inference module $\boldsymbol{R}$ in two forms, one using an LSTM (Hochreiter & Schmidhuber, 1997) and the other using a transformer (Vaswani et al., 2017) architecture. Separate networks were trained for the analogy and arithmetic tasks using each form of inference module. For each task, the grid embeddings of each stimuli selected by the DPP-S process, were provided to $\boldsymbol{R}$ as its inputs. For the arithmetic task, we also concatenated a one-hot tensor, before feeding to $\boldsymbol{R}$ that specified which computation to perform (addition or multiplication). As proposed by Hill et al. (2019), we treated both the analogy and arithmetic tasks as scoring (i.e., multiple choice) problems. For each analogy, the inference module was presented with multiple problems, each consisting of three stimuli, $A, B, C$, and a set containing $D$ (the correct completion) and six foil completions. For each instance of the arithmetic task, it was presented with two stimuli, $A, B$, and a set containing $C$ (the correct completion) and six foil completions. A linear output layer was used to generate a score for the candidate completions for each problem. Stimuli were presented sequentially for $\boldsymbol{R}$ constructed using an LSTM, and positionally coded (Kazemnejad, 2019) if it used a transformer. The seven scores (one for the correct completion and for six foil completions) were normalized using a softmax function, such that higher score would correspond to higher probablity and vice versa and the probabilities sum to 1. The inference module was trained using the task specific cross entropy loss ($\mathcal{L}_{task}$ = cross-entropy) between the softmax-normalized scores and the index for the correct completion (target).

The network that used an LSTM in the inference module had a single layer of 512 hidden units. The hidden and cell state of the LSTM was reinitialized at the start of each sequence for each candidate completion. The network that used a transformer in the inference module had 6 layers, each of which had 8 heads and a dimensionaltiy of 512. We projected the data into 128 dimensions to be more easily divisible by the number of heads, followed by layer normalization (Ba et al., 2016). We added a learnable positional encoding to the projected input sequence of attended grid code embeddings, concatenated a learned CLS (short for "classification") token (analogous to the CLS token in Devlin et al. (2018)), followed by a transformer encoder. We took the transformed value of the CLS token, and passed it to a linear layer with 1 output unit to generate a score for each candidate completion. This procedure was repeated for each candidate completion.

## 3 Related work

A body of recent computational work has shown that representations similar to grid cells can be derived using standard analytical techniques for dimensionality reduction (Dordek et al., 2016; Stachenfeld et al., 2017), as well as from error-driven learning paradigms (Cueva & Wei, 2018; Banino et al., 2018; Whittington et al., 2020; Sorscher et al., 2022). Previous work has also shown that grid cells emerge in networks trained to generalize to novel location/object combinations, and support transitive inference (Whittington et al., 2020). Here, we show that grid cells enable strong OOD generalization when coupled with the appropriate selectional

---

**Algorithm 1** Training with DPP-S

---

**Parameters:** inference module $\boldsymbol{R}$, selection weights $\boldsymbol{w}$
**Hyperparameters:** number of frequencies $N_f$, number of phases $N_p$, number of epochs optimizing DPP selection $N_{E_{DPP}}$, number of epochs optimizing task loss $N_{E_{task}}$, number of batches per epoch $N_b$
**Inputs:** covariance matrix $\boldsymbol{V}$, grid code inputs $\boldsymbol{x}$ and targets $y$ for all training problems

---

Initialize $\boldsymbol{w}$, $\boldsymbol{R}$
**for** $i = 1$ **to** $N_{E_{DPP}}$ **do**
    **for** $j = 1$ **to** $N_b$ **do**
        $\hat{F}(\boldsymbol{w}, \boldsymbol{V}, N_f, N_p) = \sum\limits_{f=1}^{N_f} \log \det(\text{diag}(\boldsymbol{w_f})(\boldsymbol{V_f} - \boldsymbol{I}) + \boldsymbol{I})$
        $\mathcal{L}_{DPP} = \text{-}\hat{F}(\boldsymbol{w}, \boldsymbol{V}, N_f, N_p)$
        Update $\boldsymbol{w}$
    **end for**
**end for**
$\hat{F}_{f \in [1, N_f]} = \log \det(\text{diag}(\boldsymbol{w_f})(\boldsymbol{V_f} - \boldsymbol{I}) + \boldsymbol{I})$
$f_{max_{DPP}} = \arg \max_{f \in [1, N_f]} \hat{F}_f$
**for** $i = 1$ **to** $N_{E_{task}}$ **do**
    **for** $j = 1$ **to** $N_b$ **do**
        $\hat{y} = \boldsymbol{R}(\boldsymbol{x}_{f=f_{max_{DPP}}})$
        $\mathcal{L}_{task} = \text{cross-entropy}(\hat{y}, y)$
        Update $\boldsymbol{R}$
    **end for**
**end for**

---

mechanism. Our proposed method is thus complementary to these previous approaches for obtaining grid cell representations from raw data.

In the field of machine learning, DPPs have been used for supervised video summarization (Gong et al., 2014), diverse recommendations (Chen et al., 2018), selecting a subset of diverse neurons to prune a neural network without hurting performance (Mariet & Sra, 2015), and diverse minibatch selection for stochastic gradient descent (Zhang et al., 2017). Recently, Mariet et al. (2019) generated approximate DPP samples by proposing an inhibitive attention mechanism based on transformer networks as a proxy for capturing the dissimilarity between feature vectors, and Perez-Beltrachini & Lapata (2021) used DPP-based attention with seq-to-seq architectures for diverse and relevant multi-document summarization. To our knowledge, however, DPPs have not previously been combined with the grid codes that we employ here, and have not been used to enable OOD generalization.

Various approaches have been proposed to prevent deep learning systems from overfitting, and enable them to egeneralize. A commonly employed technique is weight decay (Krogh & Hertz, 1992). Srivastava et al. (2014) proposed dropout, a regularization technique which reduces overfitting by randomly zeroing units from the neural network during training. Recently, Webb et al. (2020) proposed temporal context normalization (TCN) in which a normalization similar to batch normalization (Ioffe & Szegedy, 2015) was applied over the temporal dimension instead of the batch dimension. However, unlike these previous approaches, the method reported here achieves nearly perfect OOD generalization when operating over the appropriate representation, as we show in the results. Our proposed method also has the virtue of being based on a well understood, and biologically plausible, encoding scheme (grid cells).

## 4 Experiments

### 4.1 Experimental details

For each task, the sequence of stimuli for a given problem was encoded as grid codes (see Section 2.2.1), that were then modulated by DPP-S (see Section 2.2.2), and passed to the inference module $\boldsymbol{R}$ (see Section 2.2.3). We trained 3 networks using each type of inference module. For networks using an LSTM in the inference module, we trained each network for number of epochs for optimizing DPP selection $N_{E_{DPP}} = 50$, number of epochs for optimizing task loss $N_{E_{task}} = 50$, on analogy problems, and for $N_{E_{DPP}} = 500$, $N_{E_{task}} = 500$, on arithmetic problems with a batch size of 256, using the ADAM optimizer (Kingma & Ba, 2014), and a learning rate of $1e^{-3}$. For networks using a transformer in the inference module, we trained with a batch size of 128 on analogy with a learning rate of $5e^{-4}$, and on arithmetic problems with a learning rate of $5e^{-5}$. More details can be found in Appendix 7.1.

### 4.2 Comparison models

To evaluate how grid code embeddings coupled with DPP-S compares with other architectures and approaches to generalization, and the extent to which each of these components contributed to performance of the model, we compared it with several alternative models for performing the analogy and arithmetic tasks. First we compared it with the temporal context normalization (TCN) model (Webb et al., 2020) (see Section 3), but modified so as to use grid code embeddings as input. We passed the grid embedding for each input through a shared feedforward encoder which consisted of two fully connected layers with 256 units per layer. ReLU nonlinearities were used in both the layers. The final embedding was generated with a linear layer of 256 units. TCN was applied to these embeddings and then passed as a sequence for each candidate completion to the inference module. This implementation of TCN involves a learned encoder on top of the grid code embeddings, so it is closely analogous to the original TCN.

Next, we compared our model to one that used variational dropout (Gal & Ghahramani, 2016), which is shown to be more effective in generalization compared to naive dropout (Srivastava et al., 2014). We randomly sampled a dropout mask (50% dropout), zeroing out the contribution of some of the grid codes in the input to the inference module. We then use that locked dropout mask for every time step in the sequence. We also compared DPP-S to a model that had an additional penalty (L1 regularization) proportional to the absolute sum of the attention weights $w$ along with the task-specific loss. We experimented with different values of $\lambda$, which controlled the strength of the penalty relative to the cross entropy loss. We report accuracy values for $\lambda$ that achieved the best performance on the validation set. Accuracy values for various $\lambda$s are provided in the Appendix 7.7. Dropout and L1 regularization were chosen as a proxy for DPP-S and hence we used the same input data for fair comparison. Finally, we compared to a model that used the complete grid codes, i.e. no DPP-S.

## 5 Results

### 5.1 Analogy

We first present results on analogy task for two types of testing data, translation and scaling using two types of inference module, LSTM and transformer. We trained 3 networks for each method and report mean accuracy alongwith standard error of the mean. Figure 3 shows the results for the analogy task using an LSTM in the inference module. The left panel shows results for the translation regime, and the right panel shows results for the scaling regime. Both plots show accuracy on the training and validation sets, and on a series of 9 (increasingly distant) OOD generalization test regions. DPP-S (shown in blue) achieves nearly perfect accuracy on all of the test regions, considerably outperforming the other models.

For the case of translation, using all the grid codes with no DPP-S (shown in purple) led to the worst OOD generalization performance, overfitting on the training set. Locked dropout (denoted by green) and L1 regularization (denoted by red) reduced overfitting and demonstrated better OOD generalization performance than no DPP-S but still performed considerably worse than DPP-S. For the case of scaling, locked dropout

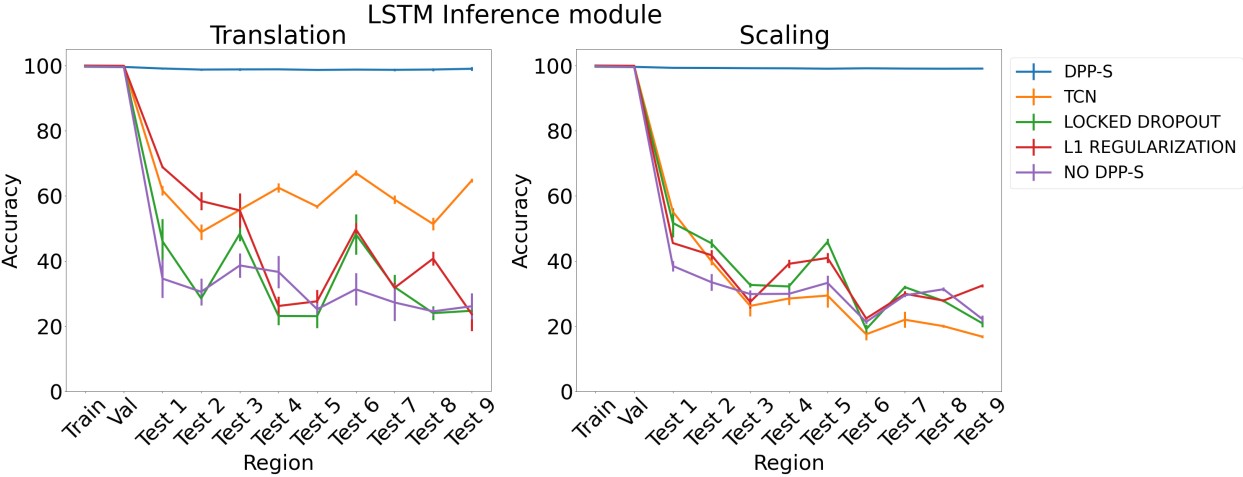

Figure 3: Results on analogy on each region for translation and scaling using LSTM in the inference module.

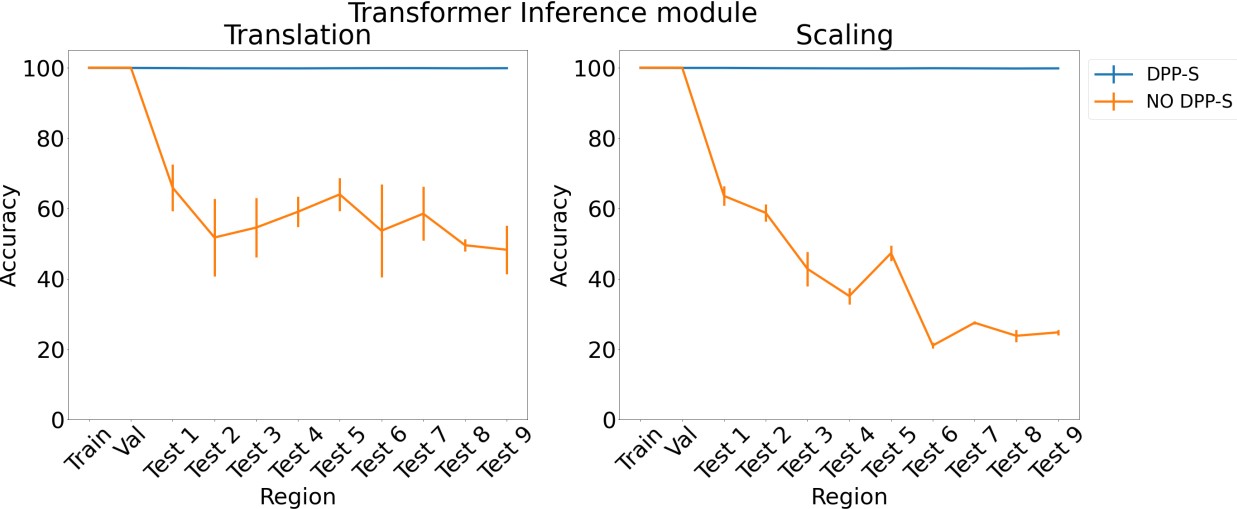

Figure 4: Results on analogy on each region for translation and scaling using transformer in the inference module.

and L1 regularization performed slightly better than TCN, achieving marginally higher test accuracy, but DPP-S still substantially outperformed all other models, with a nearly 70% improvement in average test accuracy.

To test the generality of grid embedding and DPP-S across network architectures, we also tested a transformer (Vaswani et al., 2017) in place of the LSTM in the inference module. Previous work has suggested that transformers are particularly useful for extracting structure in sequential data and has been used for OOD generalization (Saxton et al., 2019). Figure 4 shows the results for the analogy task using a transformer in the inference module. With no explicit selection (no DPP-S) over the grid codes (show in orange), the transformer did poorly on the analogies on the test regions. This suggests that simply using a more sophisticated architecture with standard forms of attention is not sufficient to enable OOD generalization based on grid codes. With DPP-S (shown in blue), the OOD generalization performance of the transformer is nearly perfect for both translation and scaling. These results also demonstrate that grid code embedding coupled with DPP-S can be exploited for OOD generalization effectively by different architectures.

## 5.2 Arithmetic

We next present results on arithmetic task for two types of operations, addition and multiplication using two types of inference module, LSTM and transformer. We trained 3 networks for each method and report mean accuracy alongwith standard error of the mean.

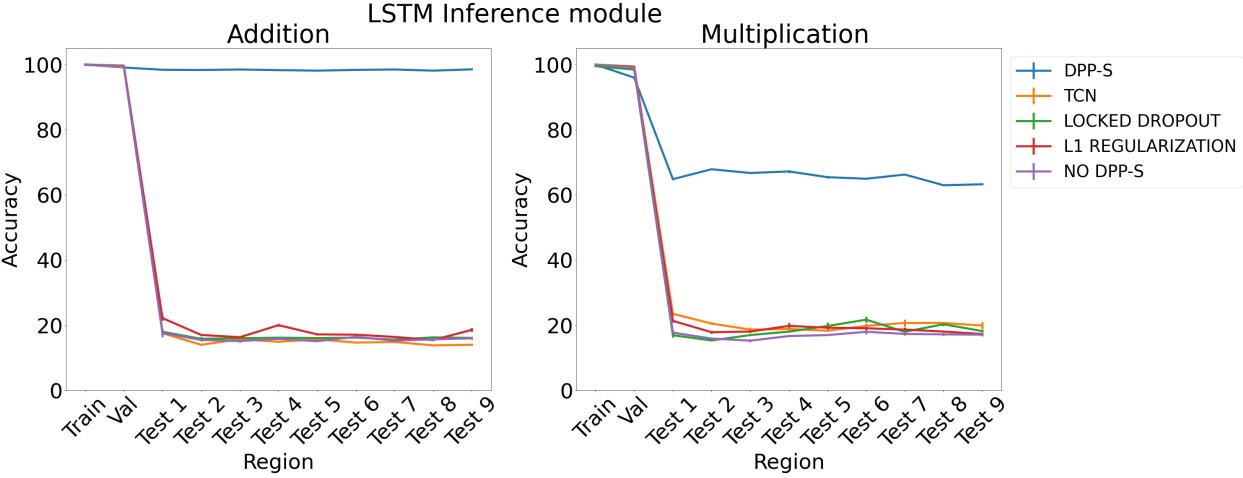

Figure 5: Results on arithmetic on each region using LSTM in the inference module.

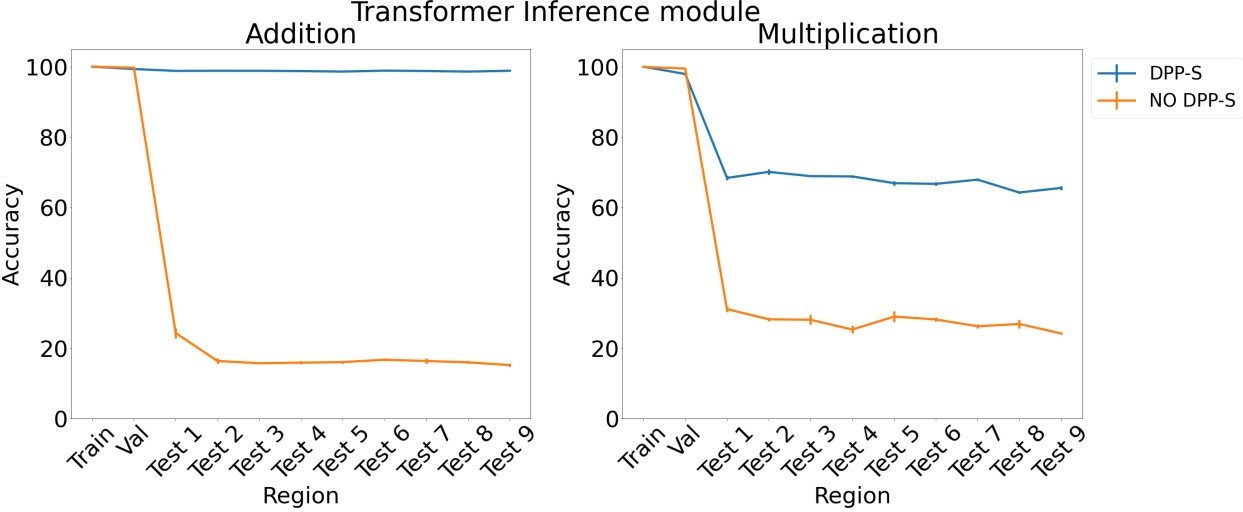

Figure 6: Results on arithmetic on each region using transformer in the inference module.

Figure 5 shows the results for arithmetic problems using an LSTM in the inference module. The left panel shows results for addition problems, and the right panel shows results for multiplication problems. DPP-S achieves higher accuracy for addition than multiplication on the test regions. However, in both cases DPP-S significantly outperforms the other models, achieving nearly perfect OOD generalization for addition, and 65% accuracy for multiplication, compared with under 20% accuracy for all the other models. We found that grid embeddings obtained after the first step in Algorithm 1 aren't able to fully preserve the relational structure for multiplication problems on the test regions (more details in Appendix 7.2), but still it affords superior capacity for OOD generalization than any of the other models. Thus, while it does not match the generalizability of a genuine algorithmic (i.e., symbolic) arithmetic function, it may be sufficient for some tasks (e.g., approximate multiplication ability in young children (Qu et al., 2021)).

Figure 6 shows the results for arithmetic problems using a transformer in the inference module. With no DPP-S over the grid codes the transformer did poorly on addition and multiplication on the test regions, achieving around 20-30% accuracy. With DPP-S, the OOD generalization performance of transformer show a pattern similar to that for analogy: it is nearly perfect for addition and, though not as good on multiplication, nevertheless show approximately 40% better performance than the transformer multiplication.

## 5.3 Ablation study

To determine the individual importance of the grid code embeddings and the DPP-S selection objective, we carried out several ablation studies. First, to evaluate the importance of grid code embeddings, we analyzed the effect of DPP-S with non-grid code embeddings, using either one-hot or smoothed one-hot embeddings with standard deviations of 1, 10, and 100, each passed through a learned feedforward encoder, which consisted of two fully connected layers with 1024 units per layer, and ReLU nonlinearities. The final embedding was generated with a fully connected layer with 1024 units and sigmoid nonlinearity. Since these embeddings don't have a frequency component, the training procedure with DPP-S consisted of only one step: minimizing the loss function $\mathcal{L} = \mathcal{L}_{task} - \lambda * \hat{F}(\boldsymbol{w}, \boldsymbol{V})$. We tried different values of $\lambda$ (0.001, 0.01, 0.1, 1, 10, 100, 1000, 10000). For each type of embedding (one-hots and smoothed one-hots with each value of $\lambda$), we trained 3 networks and report for the model that achieved best performance on the validation set. Note that, given the much higher dimensionality and therefore memory demands of embeddings based on one-hots and smoothed one-hots, we had to limit the evaluation to a subset of the total space, resulting in evaluation on only two test regions (i.e., $K \in [1,3]$).

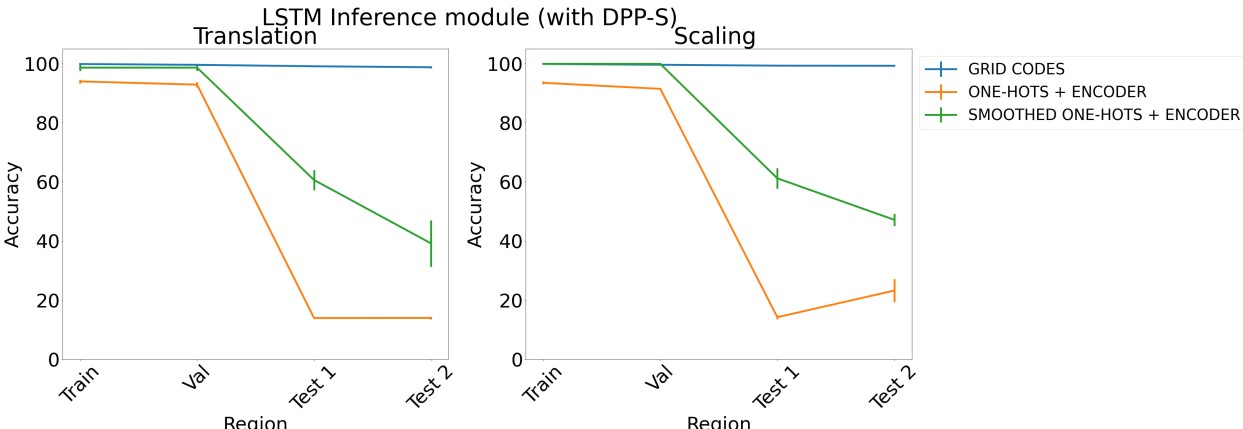

Figure 7: Results on analogy on each region using DPP-S, an LSTM in the inference module, and different embeddings (grid codes, one-hots, and smoothed one-hots passed through a learned encoder) for translation (left) and scaling (right). Each point is mean accuracy over three networks, and bars show standard error of the mean.

Figure 7 shows the results for the analogy task using an LSTM in the inference module. The average accuracy on the test regions for translation and scaling using smoothed one-hots passed through an encoder (shown in green) is nearly 30% better than simple one-hot embeddings passed through an encoder (shown in orange), but both still achieve significantly lower test accuracy than grid code embeddings which support perfect OOD generalization.

With respect to the importance of the DPP-S, we note that the simulations reported earlier show that replacing the DPP-S mechanism with either other forms of regularization (dropout and L1 regularization; see Section 4.2) or a transformer (Section 5.1 for analogy and Section 5.2 for arithmetic tasks) failed to achieve the same level of OOD generalization as the network that used DPP-S. The results using a transformer are particularly instructive, as that incorporates a powerful mechanism for learned attention, but, even when provided with grid code embeddings, failed to produce results comparable to DPP-S, suggesting that the

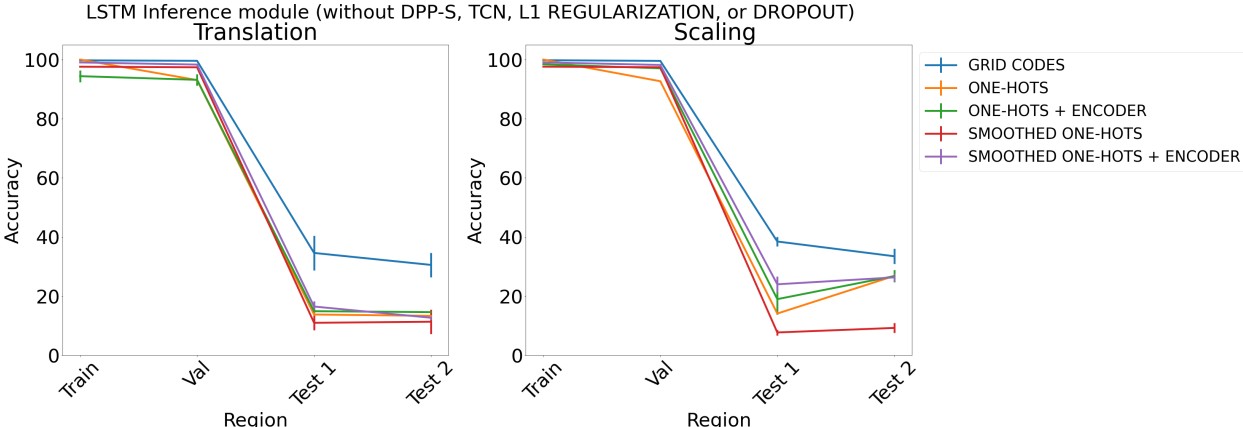

Figure 8: Results on analogy on each region using different embeddings (grid codes, and one-hots or smoothed one-hots with and without an encoder) and an LSTM in the inference module, but without DPP-S, TCN, L1 Regularization, or Dropout for translation (left) and scaling (right).

latter provides a simple but powerful form of selection objective, at least when used in conjunction with grid code embeddings.

Finally, for completeness, we also carried out a set of simulations that examined the performance of networks with various embeddings (grid codes, and one-hots or smoothed one-hots with or without a learned feedforward encoder), but no selection/attention or regularization (i.e., neither DPP-S, transformer, nor TCN, L1 Regularization, or Dropout). Figure 8 shows the results for the different embeddings. For translation (left), the average accuracy over the test regions using grid codes (shown in blue) is nearly 25% more compared to other embeddings, which all yield performance near chance ($\sim 15\%$). For scaling (right), although other embeddings achieve higher performance than chance (except smoothed one-hots), they still achieve lower test accuracy than grid codes.

## 6 Discussion and future directions

We have identified how particular properties of processing observed in the brain can be used to achieve strong OOD generalization, and introduced a two-component algorithm to promote OOD generalization in deep neural networks. The first component is a structured representation of the training data, modeled closely on known properties of grid cells – a population of cells that collectively represent abstract position using a periodic code. However, despite their intrinsic structure, we find that grid code and standard error-driven learning alone are insufficient to promote OOD generalization, and standard approaches for preventing overfitting offer only modest gains. This is addressed by the second component, using DPP-S to implement selectional regularization over the grid code. DPP-S allows only a relevant and diverse subset of cells to influence downstream computation in the inference module using the statistics of the training data. For proof of concept, we started with two cognitive tasks widely used in neuroscience (analogy and arithmetic), and showed that the combination of grid code and DPP-S promotes OOD generalization across both translation and scale when incorporated into common architectures (LSTM and transformer).

The current approach may be seen to be limited by the fact that we derive the grid codes from known properties of neural systems, rather than obtaining these codes directly from real-world data. Here, we are encouraged by the body of work providing evidence for grid-like codes in the hidden layers of neural networks in a variety of task contexts and architectures (Wei et al., 2015; Cueva & Wei, 2018; Banino et al., 2018; Whittington et al., 2020). This suggests reason for optimism that DPP-S may promote strong generalization in cases where grid-like codes naturally emerge: for example, navigation tasks (Banino et al., 2018) and reasoning by transitive inference (Whittington et al., 2020). Integrating our approach with structured representations acquired from high-dimensional, naturalistic datasets remains a critical next step which would

have significant potential for broader future practical applications. So too does application to more complex transformations beyond translation and scale, such as rotation.

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

## 7 Appendix

### 7.1 More experimental details

The size of the training region, $M$ was 100. For analogy task, we used 653216 training samples, 163304 validation samples, and 20000 testing samples for each of the nine regions. For arithmetic task, we used 80000 training samples, 20000 validation samples, and 20000 testing samples for each of the nine regions with equal number of addition and multiplication problems. We used the PyTorch library (Paszke et al., 2017) for all experiments. For each network, the training epoch that achieved the best validation accuracy was used to report performance accuracy for the training stimulus sets, validation sets (held out stimuli from the training range), and OOD generalization test sets (from regions beyond the range of the training data).

### 7.2 Why is OOD generalization performance worse for the multiplication task?

In an effort to understand why DPP-S achieved around 65% average test accuracy on multiplication compared to nearly perfect accuracy for addition and analogy task, we analyzed the distribution of the grid embeddings for the grid cells belonging to the frequency which had the maximum within-frequency determinant at the end of the first step in Algorithm 1. More specifically for $A$, $B$ and the correct answer $C$, we analyzed the

distribution of each grid cell for the training and the nine test regions. Note that since the total number of grid cells was 900 and there were nine frequencies, the dimension of the grid embeddings corresponding to $f_{max_{DPP}}$ grid cell frequency was 100. To quantify the similarity between training and the test distributions, we computed cosine distance ( 1 - cosine similarity), and averaged it over the 100 dimensions and nine test regions. We found that the average cosine distance is 5x greater for multiplication than addition problem (0.0002 for addition: 0.001 for multiplication). In this respect, grid coding does not perfectly preserve relational structure of the multiplication problem, which we would expect to limit DPP-S's OOD generalization ability in that task-domain.

## 7.3 Ablation study on choice of frequency

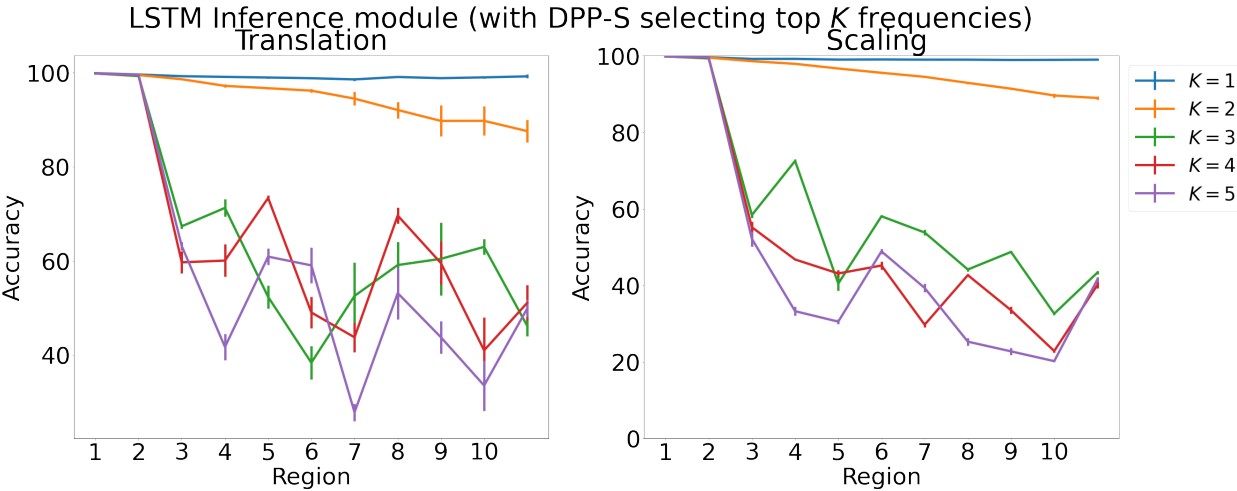

Figure 9: Results on analogy on each region using LSTM in the inference module for choosing top $K$ frequencies with $\hat{F}_f$ in Algorithm 1. Results show mean accuracy on each region averaged over 3 trained networks along with errorbar (standard error of the mean).

## 7.4 Baseline using dynamic attention across frequencies

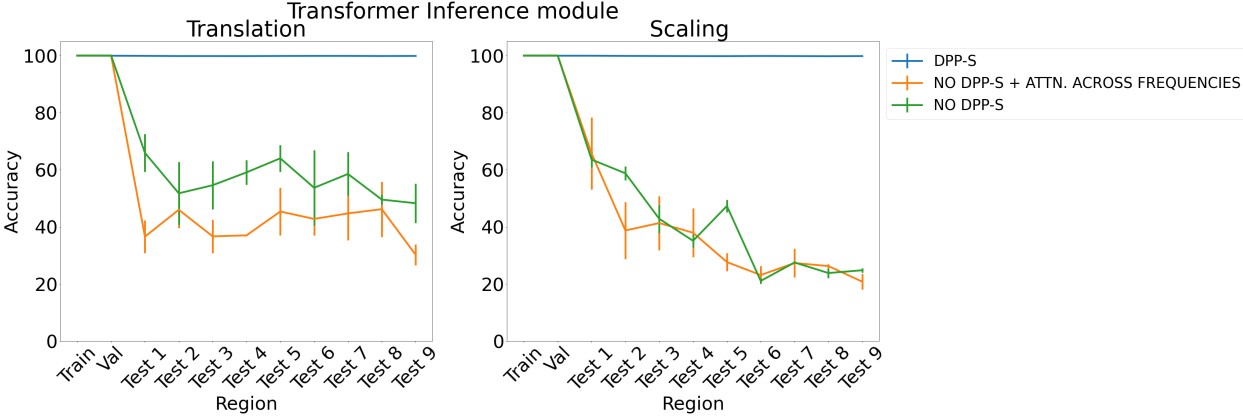

Figure 10: Results on analogy on each region for translation and scaling using transformer in the inference module.

## 7.5   Ablation study on arithmetic task

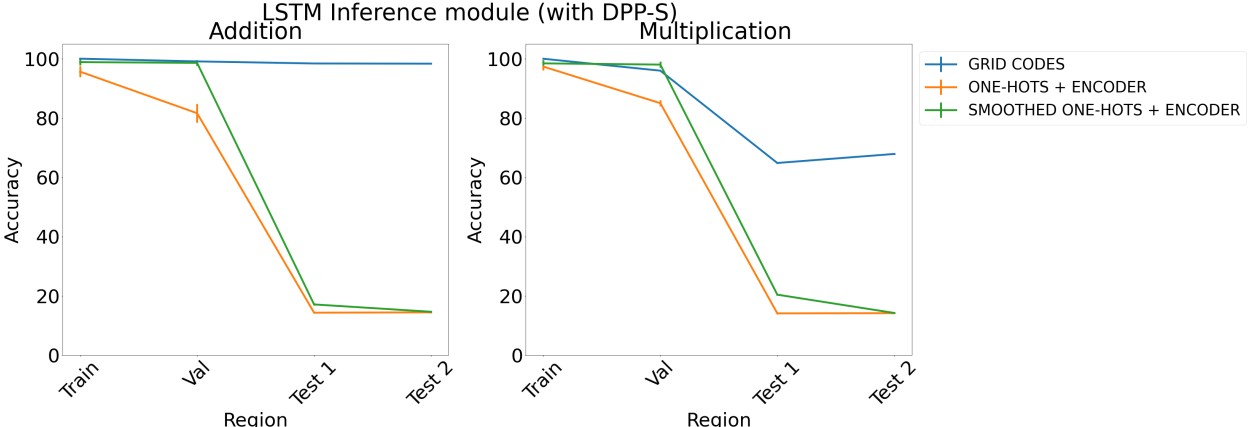

Figure 11: Results on arithmetic with different embeddings (with DPP-S) using LSTM in the inference module. Results show mean accuracy on each region averaged over 3 trained networks along with errorbar (standard error of the mean).

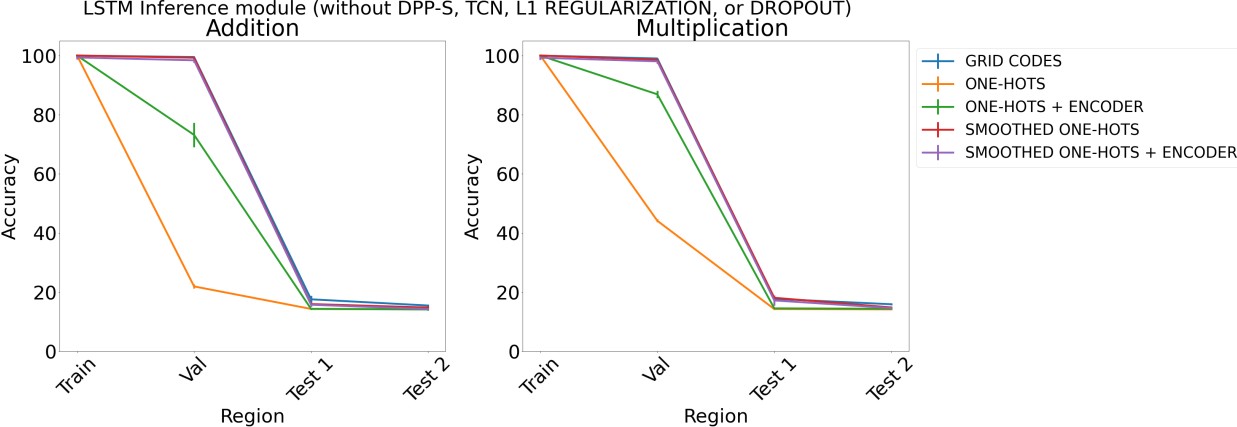

Figure 12: Results on arithmetic with different embeddings (without DPP-S, TCN, L1 Regularization, or Dropout) using LSTM in the inference module. Results show mean accuracy on each region averaged over 3 trained networks along with errorbar (standard error of the mean).

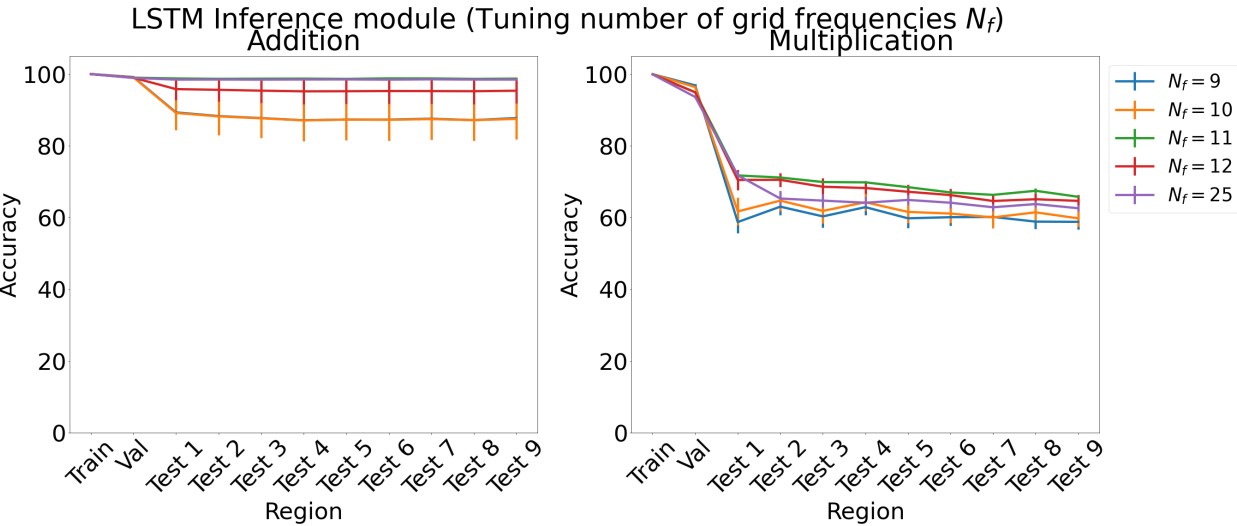

Figure 13: Results on arithmetic for increasing number of grid cell frequencies $N_f$ on each region using LSTM in the inference module. Results show mean accuracy on each region averaged over 3 trained networks along with errorbar (standard error of the mean).

### 7.6 Regression formulation

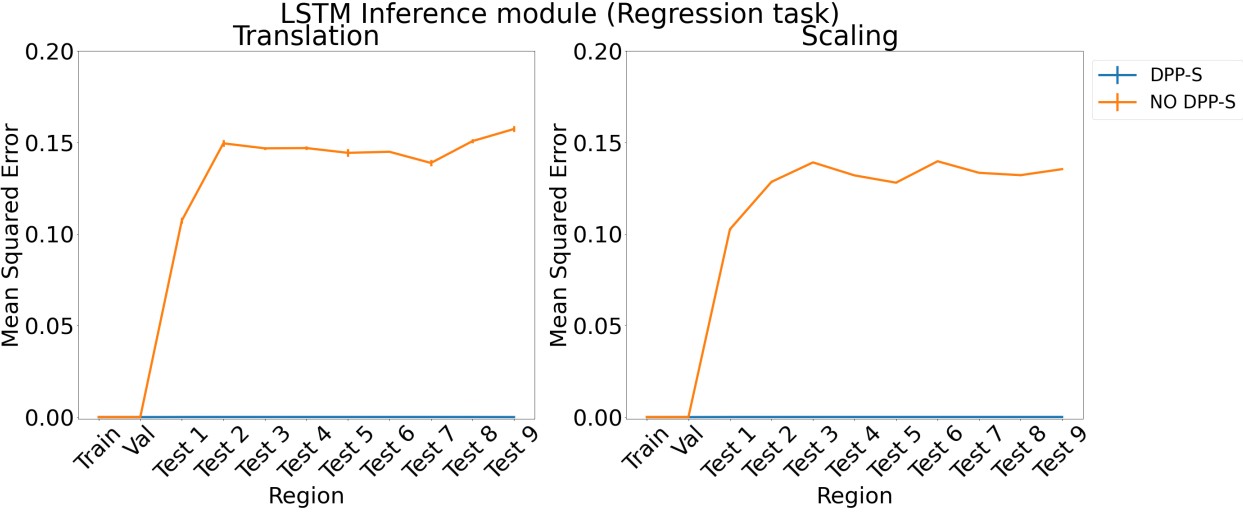

Figure 14: Results for regression on analogy using LSTM in the inference module. Results show mean squared error on each region averaged over 3 trained networks along with errorbar (standard error of the mean).

We also tried formulating the analogy and arithmetic tasks as regression instead of classification via a scoring mechanism. For DPP-S, the inference module was trained to generate the grid embeddings for grid cells belonging to the $f_{max_{DPP}}$ frequency, which had the maximum within-frequency determinant at the end of first step in Algorithm 1 for the correct completion, given as input the $f_{max_{DPP}}$ frequency grid embeddings for $A, B, C$ for the analogy task and $A, B$ for the arithmetic task. A linear layer with 100 units and sigmoid activation was used to generate the output of the inference module and was trained to minimize the mean squared error with the $f_{max_{DPP}}$ frequency grid embeddings of the correct completion. We compared DPP-S with a version that didn't use the selectional objective (no DPP-S), where the inference module was trained

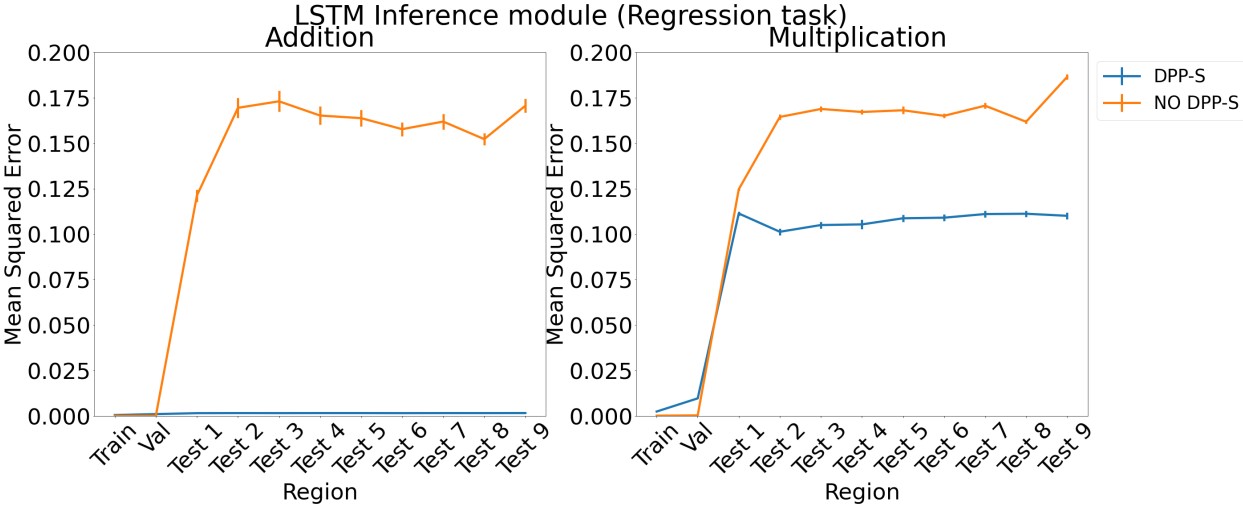

Figure 15: Results for regression on arithmetic on each region using LSTM in the inference module. Results show mean squared error on each region averaged over 3 trained networks along with errorbar (standard error of the mean).

to generate the grid embeddings for all the frequencies, but was evaluated on only the $f_{max_{DPP}}$ frequency grid embeddings for fair comparison with the DPP-S version. Figure 14 shows the results for the analogy task using an LSTM in the inference module. For both the translation (left) and scaling (right) regimes, DPP-S achieves nearly zero mean squared error on all the test regions, considerably outperforming the no DPP-S which achieves much higher error. Figure 15 shows the results for arithmetic problems using an LSTM in the inference module. For addition problems, shown on the left, DPP-S achieves nearly zero mean squared error on the test regions. For multiplication problems, shown on the right, DPP-S achieves lower mean squared error on the test regions, 0.11, compared to no DPP-S which achieves around 0.17.

## 7.7 Effect of L1 Regularization strength ($\lambda$)

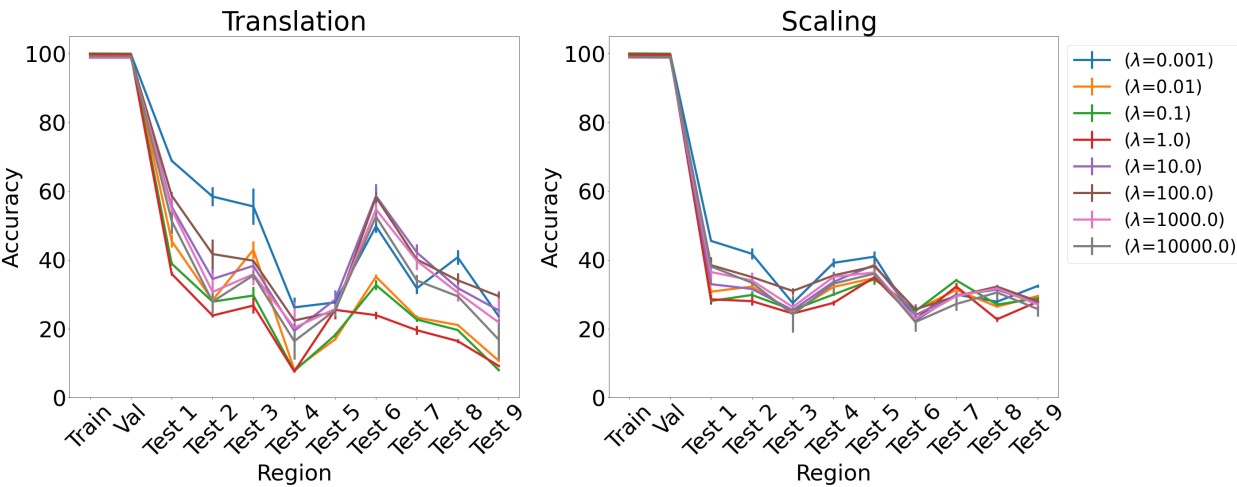

Figure 16: Results on analogy for L1 regularization for various $\lambda$s for translation and scaling using LSTM in the inference module. Results show mean accuracy on each region averaged over 3 trained networks along with errorbar (standard error of the mean).

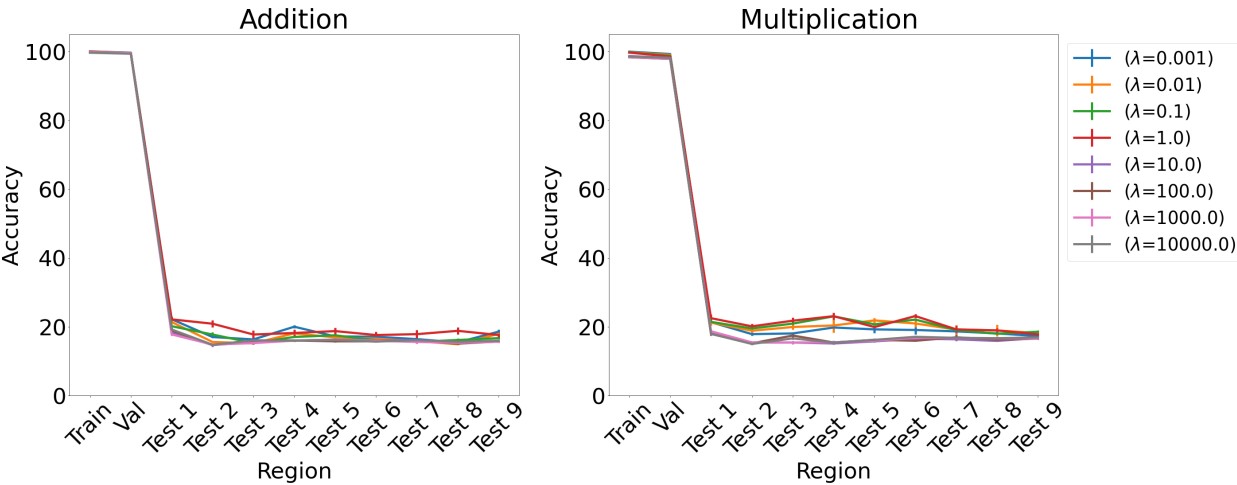

Figure 17: Results on arithmetic for L1 regularization for various $\lambda$s using LSTM in the inference module. Results show mean accuracy on each region averaged over 3 trained networks along with errorbar (standard error of the mean).

## 7.8 Ablation on DPP-S

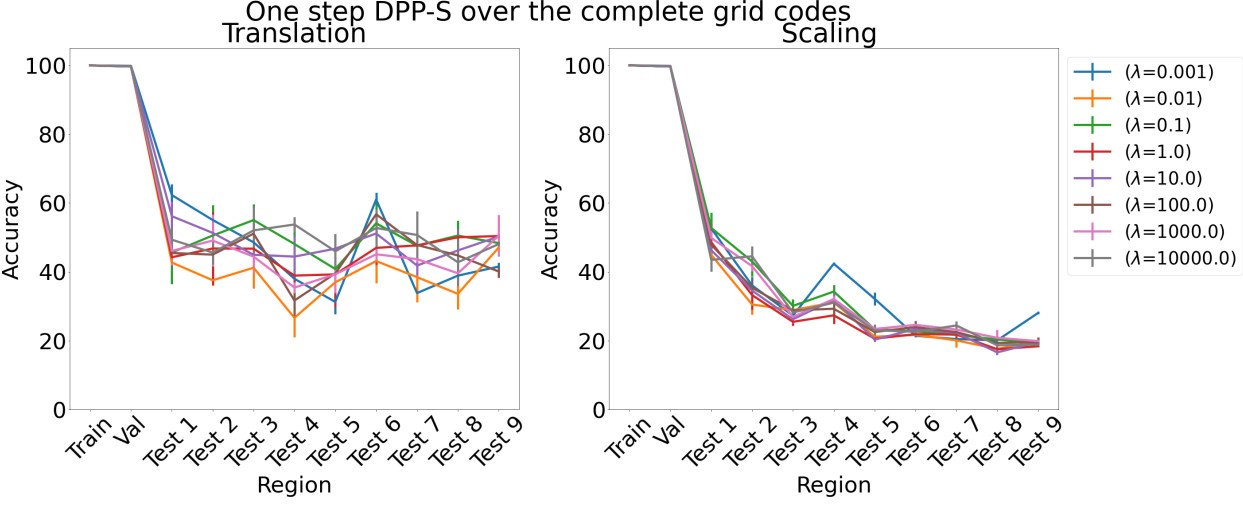

Figure 18: Results on analogy for one step DPP-S over the complete grid codes for various $\lambda$s for translation and scaling using LSTM in the inference module. Results show mean accuracy on each region averaged over 3 trained networks along with errorbar (standard error of the mean).

The proposed DPP-S method (Algorithm 1) consists of two steps with $\mathcal{L}_{DPP}$ in the first step and $\mathcal{L}_{task}$ in the second step. We considered two ablation experiments which consists of a single step. In one case we maximized the objective function, $\hat{F}(\boldsymbol{w}, \boldsymbol{V}) = \log \det(diag(\boldsymbol{w})(\boldsymbol{V} - \boldsymbol{I}) + \boldsymbol{I})$, over the complete grid codes (instead of summing $\hat{F}$ corresponding to grid codes from each frequency independently as done in the first step of Algorithm 1), using stochastic gradient ascent, along with minimizing $\mathcal{L}_{task}$, which would use all the attended grid codes (instead of using $f_{max_{DPP}}$ frequency grid codes as done in the second step of Algorithm 1). So total loss, $\mathcal{L} = \mathcal{L}_{task} - \lambda * \hat{F}(\boldsymbol{w}, \boldsymbol{V})$. We refer to this ablation experiment as one step DPP-S over the complete grid codes. The results on analogy for this ablation experiment is shown in Figure 18. We see that the accuracy on test analogies for translation for various $\lambda$s are around 30-60%, and for scaling around 20-40%, which is much lower than the nearly perfect accuracy achieved by the proposed DPP-S method. In the

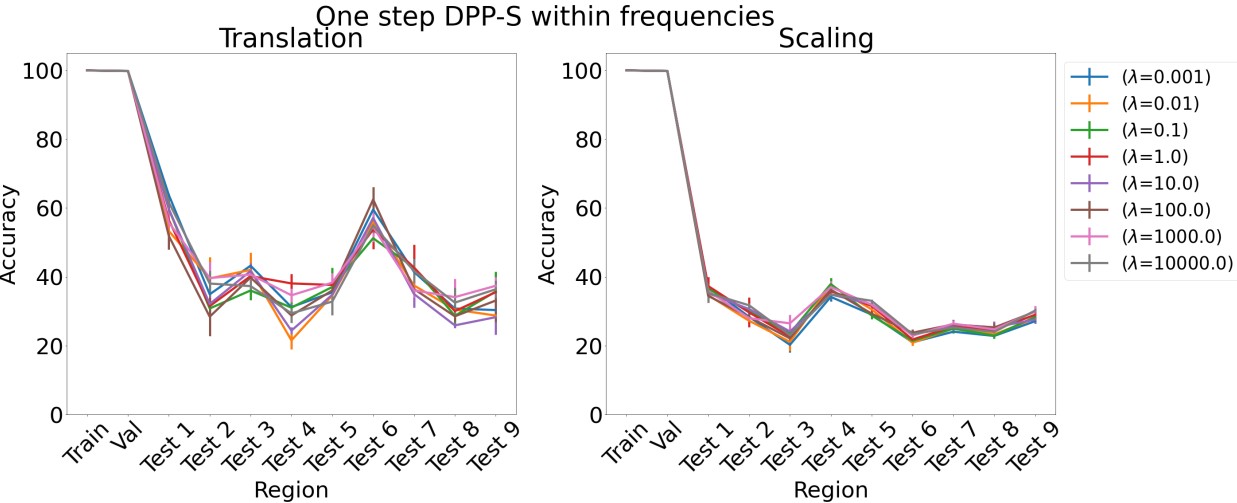

Figure 19: Results on analogy for one step DPP-S within frequencies for various $\lambda$s for translation and scaling using LSTM in the inference module. Results show mean accuracy on each region averaged over 3 trained networks along with errorbar (standard error of the mean).

other case we maximized the objective function $\hat{F}(\boldsymbol{w}, \boldsymbol{V}, N_f, N_p) = \sum_{f=1}^{N_f} \log \det(diag(\boldsymbol{w_f})(\boldsymbol{V_f} - \boldsymbol{I}) + \boldsymbol{I})$, using stochastic gradient ascent, which is same as $\mathcal{L}_{DPP}$ in the first step of Algorithm 1, along with minimizing $\mathcal{L}_{task}$, which would use all the attended grid codes. So total loss, $\mathcal{L} = \mathcal{L}_{task} - \lambda * \hat{F}(\boldsymbol{w}, \boldsymbol{V})$. We refer to this ablation experiment as one step DPP-S within frequencies. As shown in Figure 19, the accuracy on test analogies for both translation and scaling for various $\lambda$s are in a similar range to one step DPP-S over the complete grid codes, and is much lower than the nearly perfect accuracy achieved by the proposed DPP-S method.

