# OpenReview forum: "Determinantal Point Process Selection Over Grid Codes Supports Out of Distribution Generalization"
_TMLR — Rejected by TMLR_

### Review · Reviewer_N22L · 2023-02-03

**Summary Of Contributions:**

The paper shows that an attention mechanism based on a Determinantal Point Process (DPP) over grid-cell embeddings can help downstream models to extrapolate on toy reasoning tasks. While neither of the two main components of the resulting method (i.e. grid cells codes and DPP attention) is new and has been explored in prior work, the novelty of the method consists of: (1) the way in which the two are combined and (2) they way they can improve generalisation performance.

**Audience:**

Yes

**Claims And Evidence:**

Yes

**Requested Changes:**

- As mentioned above, I think the paper would benefit from a slight change/clarification of narrative, focusing on how attention can aid generalisation in grid-cell representations, which is (after all) also what the title claims. I think from this perspective, this is a good paper. On the other hand, the algorithm and evaluation clearly do not demonstrate (in the current stage) any utility for practical applications and this direction is not sufficiently supported.
- Nit: The definition of extrapolation from the abstract as "successful performance on test examples that lie outside the convex hull
of the training set" is a bit too simplistic. In a very high-dimensional space, there might be data points that are very hard to extrapolate *inside* the convex hull of the training data. What is important is whether the data is in or out of distribution.

**Strengths And Weaknesses:**

***Strengths***

- The paper combines two of the most important ideas from neuroscience, grid cells and attention, in a sensible way. Effectively, DPP-based attention is used to select the right scale for the problem by picking only the embeddings of the appropriate frequency.
- The experiments convincingly show that the DPP attention promotes extrapolation in the considered tasks.
- The ablation study over regularisation methods demonstrates that the DPP-attention is far superior in these tasks to the usual regularisation methods (e.g. dropout).
- The experiments provide some evidence that the good extrapolation effect is to some degree present irrespective of the downstream model by validating the results both on LSTMs and Transformers.

***Weaknesses***

- My main point is that from the narrative of the paper, the intended purpose of the proposed algorithm is unclear to me. In my opinion, it is nice as a toy computational model to potentially evaluate how generalisation in the human brain might work: i.e. to empirically check how attention might aid generalisation in grid cell representations. On the other hand, it is clear that (at least in the current stage) this is not a practical algorithm ready to be deployed in some real-world applications or even some more realistic benchmarks. I think it would be good if the authors could clarify (both here but also in the paper) which one of the two is it.
- Related to the above, if the intended purpose is to propose a practical algorithm, I believe that the experiments showing that the grid-cell embeddings are necessary are not very convincing. The alternative representations tested (e.g. one-hot) are too simple and (unlike grid-cell code) lack any multi-scale component. Could for instance a multi-scale one-hot encoding or some Fourier representation help? At the opposite end, what about simply using the coordinates of the vectors as the raw features? Again, on the other hand, if the goal is to evaluate how attention might work together with grid cells in the human brain, then grid-cells are by default our starting point and I do not see the need to evaluate against other representations.
- Could you also clarify how the one-hot encodings are exactly produced? I assumed these are based on a discretisation of the plane. Is that correct?
- I am not sure why the reasoning tasks are framed as a classification instead of regression. The classification formulation with 6 wrong answers makes the tasks significantly easier. Is it because the method does not work so well otherwise?
- As the authors mention, the representations are hand-crafted (i.e. not learned) and the range of transformations of the data that were considered are also very simple (scaling and translations).

---

> ### Comment · Reviewer_6GAB · 2023-02-18
> **It is not really *attention***
>
> The authors use the word *attention* a lot in their paper, but it does not match a core property of attention, be it in brains or in deep learning: attention makes a context-dependent selection. Instead, the proposed algorithm statically selects (once and for all) a subset of the inputs and then ignores the others. Attention should by dynamic, input-dependent.

---

> > ### Comment · Reviewer_N22L · 2023-02-22
> > **Re *attention***
> >
> > Thank you for pointing this out! My original understanding was that at test time, $f_{max}$ would depend on the test task, but as the authors also confirmed in their response, there is no context-dependence and the $f_{max}$ obtained at training time is used. Perhaps this could be clarified by also providing pseudocode for how the model works at test time. So far, only code for the training procedure is provided.
> >
> > Based on this clarification, I have a few more questions & experimental suggestions for the authors:
> >
> > - What happens as you pass to $R$ not only features corresponding to ${f_{max}}_{DDP}$ but features corresponding to the TopK (with $K$ a hyperparameter) frequencies? The paper seems to assume that only the top frequency is useful but, in principle, one could retrieve some useful information from multiple (or all) frequencies. I think this would be a reasonable ablation as the choice of the top frequency is a bit arbitrary.
> > - Could (dynamic/"true") attention automatically learn to select information from the useful frequencies? As far as I can see, this has not been examined and it seems an essential baseline. The Transformer in Figure 6 only attends over the inputs but not across the different frequency vectors $\mathbf{x}_f$, where the *static* "attention" is applied. So instead of computing the argmax in the pseudocode, attention could do a soft argmax.

---

> ### Author Response · Authors · 2023-02-19
> **Clarification for Reviewer N22L**
>
> We thank the reviewer for their helpful suggestions and comments. We are in the process of addressing the requested changes, and will get back with a more detailed response soon. In the meantime, we would like to clarify that the purpose of the paper is conceptual, demonstrating how certain principles associated with neural processing can, when appropriately combined, achieve strong extrapolative generalization. We do not view the present work as making an immediate practical contribution, but we do believe that the principles investigated in this work have significant potential for future practical applications. In order to extend our proposed approach to more practical domains, methods would need to be developed to extract grid-like codes from high-dimensional real-world data, but a number of recent results ([1,2,3]) suggest that this should be possible. We will revise the paper so as to clarify the framing of our contribution.
>
> With regard to the specific suggestion that we consider a multi-scale one-hot encoding, while this is an interesting idea, it would be subject to an exaggerated form of the constraints faced by simpler location-specific codings (e.g., simple one-hot encodings, as well as “smoothed one-hot encodings”), which is an exponential increase in dimensionality and concomitant representational demands as the size of the space increases, posing a practical problem regarding memory constraints for simulation (and presumably a similar one in an actual neural system). Specifically, for one-hot encodings, each input is represented by a $N^2$ dimensional vector where the index corresponding to the position of the input on a flattened $N \times N$ grid has a value of 1 and the rest are zeros. We had to significantly cut the space evaluating on only two test regions for one-hot encodings to address memory limitations, and adding more scales would increase the dimension by a factor of the number scales, and we won’t be able to evaluate on a single test region even with a batch size of 1 on a GPU with 16GB memory. This also shows the usefulness of a reasonable low-dimensional representation like grid codes.  In contrast, grid codes do not suffer from this problem, with the number of cells scaling logarithmically with the level of resolution required by the system.
>
>
> [1] - Cueva, C.J. and Wei, X.X., 2018. Emergence of grid-like representations by training recurrent neural networks to perform spatial localization. arXiv preprint arXiv:1803.07770.
>
> [2] - Banino, A., Barry, C., Uria, B., Blundell, C., Lillicrap, T., Mirowski, P., Pritzel, A., Chadwick, M.J., Degris, T., Modayil, J. and Wayne, G., 2018. Vector-based navigation using grid-like representations in artificial agents. Nature, 557(7705), pp.429-433.
>
> [3] - Whittington, J.C., Muller, T.H., Mark, S., Chen, G., Barry, C., Burgess, N. and Behrens, T.E., 2020. The Tolman-Eichenbaum machine: unifying space and relational memory through generalization in the hippocampal formation. Cell, 183(5), pp.1249-1263.

---

> ### Author Response · Authors · 2023-03-01
> **Response to reviewer N22L**
>
> We thank the reviewer for their insightful feedback and suggestions. We have uploaded a revised version of the manuscript. Below we include a detailed response to the reviewers' comments.
>
> -*if the intended purpose is to propose a practical algorithm, I believe that the experiments showing that the grid-cell embeddings are necessary are not very convincing. The alternative representations tested (e.g. one-hot) are too simple and (unlike grid-cell code) lack any multi-scale component. Could for instance a multi-scale one-hot encoding or some Fourier representation help? At the opposite end, what about simply using the coordinates of the vectors as the raw features? Again, on the other hand, if the goal is to evaluate how attention might work together with grid cells in the human brain, then grid-cells are by default our starting point and I do not see the need to evaluate against other representations.*
>
> -*As mentioned above, I think the paper would benefit from a slight change/clarification of narrative, focusing on how attention can aid generalisation in grid-cell representations, which is (after all) also what the title claims. I think from this perspective, this is a good paper. On the other hand, the algorithm and evaluation clearly do not demonstrate (in the current stage) any utility for practical applications and this direction is not sufficiently supported.*
>
> We appreciate the concerns raised by the reviewer, which have motivated us to more clearly articulate the purpose of the paper. This paper focuses on the challenge of understanding how the capacity for generalization arises in natural neural systems, which may inspire future work addressing how this can be applied in artificial neural networks.  Specifically the purpose of the paper is to demonstrate how particular properties of processing observed in the brain – in particular, the use of grid cell embeddings –  can be used to achieve strong out-of-distribution generalization, when combined with a generic selective filter over those embeddings. For proof of concept, we focused on tasks involving two dimensional stimuli , which have similarly been the focus of work in neuroscience (i.e., spatial location), and therefore serve as a reasonable starting point for examining the properties of interest in simulated networks. Based on our findings, we believe that the mechanisms  we have identified should apply readily to higher dimensional stimuli, and therefore have significant potential for broader future practical applications. However,  in order to extend this  approach to more practical domains (e.g. involving end-to-end training), methods would need to be developed to extract grid-like codes from such high-dimensional real-world data.  This remains a current challenge, although several recent results (Wei et al., 2015; Cueva & Wei, 2018; Banino et al., 2018;Whittington et al., 2020) suggest that this should be possible. We have revised the paper so as to clarify the framing of our contribution and potential for more practical applications. The relevant sentences are:
>
>
> ### From the Abstract
>
> “Here, we identify how certain properties of processing in the brain can be used to achieve strong OOD generalization and offer a two-part algorithm that improves OOD generalization performance of artificial neural networks on multiple tasks widely used in neuroscience to demonstrate proof of concept.”
>
> ### From the Introduction
>
> “ Here, we consider two cognitive problems that are widely used in neuroscience and often require a capacity for OOD generalization: a) analogy and b) arithmetic. What enables the human brain to successfully generalize on these tasks, and how might we better realize that ability in deep learning systems?”
>
> “Taken together, the representational and selection mechanisms outlined above define a two-component framework for promoting OOD generalization, by minimizing task-specific error subject to: i) embeddings that encode relational structure among the data (grid cells), and ii) selection of those embeddings that maximize the "volume" of the representational space that is covered, while minimizing redundancy (DPP-S). Below, we demonstrate proof of concept by showing that these mechanisms allow artificial neural networks to learn representations that support OOD generalization on two cognitive tasks which have similarly been the focus in neuroscience and therefore serve as a reasonable starting point for examining the properties of interest in these networks.”

---

> > ### Author Response · Authors · 2023-03-01
> > **Response to reviewer N22L (Part 2)**
> >
> > ### From the Discussion and future directions
> >
> > “We have identified how particular properties of processing observed in the brain can be used to achieve strong OOD generalization, and introduced a two-component algorithm to promote OOD generalization in deep neural networks.”
> >
> > “For proof of concept, we started with two cognitive tasks widely used in neuroscience (analogy and arithmetic), and showed that the combination of grid code and DPP-S promotes OOD generalization across both translation and scale when incorporated into common architectures (LSTM and transformer).”
> >
> > “The current approach may be seen to be limited by the fact that we derive the grid codes from known properties of neural systems, rather than obtaining these codes directly from real-world data. Here, we are encouraged by the body of work providing evidence for grid-like codes in the hidden layers of neural networks in a variety of task contexts and architectures (Wei et al., 2015; Cueva & Wei, 2018; Banino et al., 2018;Whittington et al., 2020). This suggests reason for optimism that DPP-S may promote strong generalization in cases where grid-like codes naturally emerge: for example, navigation tasks (Banino et al., 2018) and reasoning by transitive inference (Whittington et al., 2020). Integrating our approach with structured representations acquired from high-dimensional, naturalistic datasets remains a critical next step which would have significant potential for broader future practical applications. So too does application to more complex transformations beyond translation and scale, such as rotation.”
> >
> >
> > References:
> >
> >
> > Wei, X.X., Prentice, J. and Balasubramanian, V., 2015. A principle of economy predicts the functional architecture of grid cells. Elife, 4, p.e08362.
> >
> > Cueva, C.J. and Wei, X.X., 2018. Emergence of grid-like representations by training recurrent neural networks to perform spatial localization. arXiv preprint arXiv:1803.07770.
> >
> > Banino, A., Barry, C., Uria, B., Blundell, C., Lillicrap, T., Mirowski, P., Pritzel, A., Chadwick, M.J., Degris, T., Modayil, J. and Wayne, G., 2018. Vector-based navigation using grid-like representations in artificial agents. Nature, 557(7705), pp.429-433.
> >
> > Whittington, J.C., Muller, T.H., Mark, S., Chen, G., Barry, C., Burgess, N. and Behrens, T.E., 2020. The Tolman-Eichenbaum machine: unifying space and relational memory through generalization in the hippocampal formation. Cell, 183(5), pp.1249-1263.
> >
> > With regard to alternative representations, we agree that one-hot encoding are too simple and do not  include a multi-scale component, as do grid codes. To address the reviewer’s suggestion that we include a comparison to grid codes that includes a multi-scale component, we have now evaluated a multi scale one-hot encoding in which there are four scales. The first scale is the standard one-hot encoding in which the index corresponding to the position of the input point on a flattened grid of NxN locations has a value of 1 and the rest are zeros. For the second, third and fourth scales 1%, 5%, and 10% of the indices in the neighbourdhood of the index corresponding to the position of the input point on a flattened grid of NxN locations have values of 1 and the rest are zeros respectively.  These additional scales pose substantial additional memory demands, above and beyond the already large demands for one-hot encoding of indvidual locations. To make simulations using these representations tractable on a 80GB A100 GPU, we limited evaluation to only one training and test region. The test accuracy obtained is around 93% and 89% lower for translation and scaling respectively than our proposed DPP-S method.  We also tried using the coordinates of the vectors after normalizing them to the range [0,1], followed by applying temporal context normalization over the  sequence of inputs. Although the average test accuracy on the analogy task is very similar to our proposed DPP-S method, it achieves around 80% and 5% lower test accuracy on addition and multiplication respectively.
> >
> > With respect to the need to evaluate against other representations, our proposed method consists of two parts, grid cells which provide a representation of relational structure in which relations are preserved across transformations and a selectional objective to maximize variance and minimize correlation. Our experiments demonstrate that both parts are important to achieve significant out-of-distribution generalization on analogy and arithmetic tasks by showing that grid cells alone, without the selectional objective and other representations with and without the selectional objective aren’t able to generalize out-of-distribution.

---

> > > ### Author Response · Authors · 2023-03-01
> > > **Response to reviewer N22L (Part 3)**
> > >
> > > -*Could you also clarify how the one-hot encodings are exactly produced? I assumed these are based on a discretisation of the plane. Is that correct?*
> > >
> > > For one-hot encodings each input 2d point $ \in \mathbb{Z}^{2}$ corresponding to the task is represented by a ${KM}^2$ dimensional vector where the index corresponding to the position of the point on a flattened grid of $KM \times KM$ locations has a value of 1 and the rest are zeros. Here $M$ denotes the size of the training region on either dimension and $K-1$ denotes the number of test regions.
> > >
> > >
> > > -*I am not sure why the reasoning tasks are framed as a classification instead of regression. The classification formulation with 6 wrong answers makes the tasks significantly easier. Is it because the method does not work so well otherwise?*
> > >
> > > We thank the reviewer for the question. We have formulated the analogy and arithmetic tasks as regression and have included the results in Appendix, section 7.6. Figure 14 shows the results on analogy task where DPP-S achieves nearly zero mean squared error when trained to generate $f_{max_{DPP}}$ frequency grid embeddings for the correct completion compared to no DPP-S which is trained to generate grid embeddings for all the frequencies but evaluated on only $f_{max_{DPP}}$ frequency grid embeddings for fair comparison. Figure 15 shows the results on arithmetic task where for addition, DPP-S achieves nearly zero mean squared error on the test regions and for multiplication, DPP-S achieves lower mean squared error on the test regions, $0.11$, compared to no DPP-S which achieves around $0.17$.
> > >
> > > -*As the authors mention, the representations are hand-crafted (i.e. not learned) and the range of transformations of the data that were considered are also very simple (scaling and translations).*
> > >
> > > We agree that the current version of the method is limited with respect to the fact that the representations are not learned.  Learning grid cell embeddings remains a challenge, though some recent results (Wei et al., 2015; Cueva & Wei, 2018; Banino et al., 2018;Whittington et al., 2020) suggest that this should be possible. Evaluating these approaches, and integrating them into the work we present here is an important direction for future work, as is the testing of the these methods on more complex transformations (e.g. rotation).
> > >
> > > References:
> > >
> > >
> > > Wei, X.X., Prentice, J. and Balasubramanian, V., 2015. A principle of economy predicts the functional architecture of grid cells. Elife, 4, p.e08362.
> > >
> > > Cueva, C.J. and Wei, X.X., 2018. Emergence of grid-like representations by training recurrent neural networks to perform spatial localization. arXiv preprint arXiv:1803.07770.
> > >
> > > Banino, A., Barry, C., Uria, B., Blundell, C., Lillicrap, T., Mirowski, P., Pritzel, A., Chadwick, M.J., Degris, T., Modayil, J. and Wayne, G., 2018. Vector-based navigation using grid-like representations in artificial agents. Nature, 557(7705), pp.429-433.
> > >
> > > Whittington, J.C., Muller, T.H., Mark, S., Chen, G., Barry, C., Burgess, N. and Behrens, T.E., 2020. The Tolman-Eichenbaum machine: unifying space and relational memory through generalization in the hippocampal formation. Cell, 183(5), pp.1249-1263.
> > >
> > >
> > > -*Nit: The definition of extrapolation from the abstract as "successful performance on test examples that lie outside the convex hull of the training set" is a bit too simplistic. In a very high-dimensional space, there might be data points that are very hard to extrapolate inside the convex hull of the training data. What is important is whether the data is in or out of distribution.*
> > >
> > > We thank the reviewer for raising this point. We agree that the definition of extrapolation as successful performance on test examples that lie outside the convex hull of the training set is too simplistic, and that a more challenging (and naturalistic) interpretation of extrapolation is the ability to generalize “outside the range” of the training data.  Accordingly, we have changed extrapolation to out-of-distribution generalization in the revised version of the manuscript.

---

> > > > ### Author Response · Authors · 2023-03-01
> > > > **Response to reviewer N22L (Part 4)**
> > > >
> > > > -*What happens as you pass to $R$ not only features corresponding to  $f_{{max}_{DPP}}$ but features corresponding to the TopK (with $K$ a hyperparameter) frequencies? The paper seems to assume that only the top frequency is useful but, in principle, one could retrieve some useful information from multiple (or all) frequencies. I think this would be a reasonable ablation as the choice of the top frequency is a bit arbitrary.*
> > > >
> > > > We thank the reviewer for suggesting this additional ablation experiment. We have performed the experiment and have included the results in the Appendix, section 7.3. We observed that as we pass to $R$ features corresponding to top $K$ frequencies based on $\hat{F}_f$ in Algorithm 1, the average test accuracy on the analogy task goes down as the value of $K$ increases as shown in Figure 9. There is a 10% reduction in accuracy for the farther test regions when increasing $K$ from 1 to 2, and more than 40% reduction for values of $K$ greater than 2. This shows that only the top frequency is useful and including more frequencies hurts out-of-distribution generalization performance.
> > > >
> > > > -*Could (dynamic/"true") attention automatically learn to select information from the useful frequencies? As far as I can see, this has not been examined and it seems an essential baseline. The Transformer in Figure 6 only attends over the inputs but not across the different frequency vectors $x_f$, where the static "attention" is applied. So instead of computing the argmax in the pseudocode, attention could do a soft argmax.*
> > > >
> > > > We thank the reviewer for suggesting this additional baseline. We have performed the experiment and have included the results in the Appendix, section 7.4. For this additional baseline, we trained the transformer with a learnable softmax-normalized parameter across different frequencies, but shared for vectors belonging to the same frequency. The average test accuracy is lower for translation and similar for scaling than the standard transformer (NO DPP-S) as shown in Figure 10. This suggests that using a softmax based attention mechanism for selecting information from different frequencies doesn’t help with OOD generalization.

---

### Review · Reviewer_6GAB · 2023-02-08

**Summary Of Contributions:**

The authors present a novel architecture and regularizer for supervised learning in 2-dimensional input space. The low-dim input is represented with a very large number (900) of frequency/phase units similar to positional encoding of Transformers and inspired by grid codes found in brains. A determinantal point process formula is used to define a regularizer that makes the network only use a small but diverse subset of the grid units. Experiments with OOD generalization evaluation are performed on toy 2-D tasks showing a very significant advantage of the proposed approach against an L1 regularizer over the use of the grid units, dropout and a variant of batch normalization (TCN).

**Audience:**

Yes

**Broader Impact Concerns:**

None that I can see.

**Claims And Evidence:**

No

**Requested Changes:**

The requested changes reflect the above list of weaknesses.

* The method should be validated on tasks that exist in the ML literature (where others have done their best to get a good result) and should not be limited to so low-dimensional data, for relevance to the TMLR audience.
* The terms 'extrapolation', 'attention' and 'reasoning' should be eliminated or carefully reworded (paying attention to the above concerns).
* Places that need clarification, in order of appearance:
  - 2nd par. of intro, points (a) and (b). What is meant by "structured nature of representations" and "representational volume" is not yet clear at this point in the paper.
  - p. 2, 1st par., "invariant to translation within frequency and invariant to scale across frequencies" only becomes clear later, but the reader hasn't read those things yet.
  - p. 2, 2nd par., "within context variance ("relevance")" is not clear at this point
  - p. 5, 1st par., "conceptual states" is too vague, and seems to go against common sense that concepts live in a high-dimensional space, not in a 2-D space (except for the very specific concept of spatial location!)
  - p. 5, Eq. 2: F is not defined/or explained (what are its dimensions?), nor is A, nor is A_offset.
  - section 2.2.2 (DPP-A): the connection between the literature on DPP and the proposed architecture is too vague. Why is a such a selection of grid points expected to be useful? the questions below about algorithm 1 should be answered here.
  - algorithm 1: why is this choice of frequency f a good thing? why do we even want to select a frequency? (this should be explained in the text)


**Strengths And Weaknesses:**

Strengths:
* taking inspiration from neuroscience (for the grid units)
* an original use of DPP ideas to regularize some of the neural network weights
* the proposed method generalized much better OOD than the other ones tested

Weaknesses:
* 2-D input is not of much interest in ML
* although the notion of extrapolation is meaningful in 2-D, it isn't in high dimension (pretty much any randomly chosen test point is outside the convex set of the training examples)
* the experiments are not convincing because (a) of the toy and low-dim nature of the task (b) because there are no comparisons on previously published tasks (where other groups of researchers would have done their best) (c) in some cases the results seem too good (100% test accuracy), which suggests either bugs or that the tasks are way too easy in some sense.
* clarity needs improvements in several places, including regarding the reason why the DPP formula would be helpful.
* using the term "attention" seems inappropriate because the selection is unconditional; it is really *statically* modulating down some of the input weights. Attention (as the word is used both in brain sciences and in deep learning) is dynamic and depends on context.
* using the term "reasoning" does not seem appropriate either; reasoning should be about coherently or logically composing pieces of knowledge to arrive at a conclusion in a way similar to how we think. What the network shown does is more like a standard supervised learning task, which is generally not recognized as performing reasoning (or only in a very weak sense).

---

> ### Author Response · Authors · 2023-02-19
> **Terminology clarification for Reviewer 6GAB**
>
> We thank the reviewer for their helpful suggestions and comments.  We are in the process of addressing the requested changes, and will get back with a more detailed response soon. In the meantime, we would like to address some of the concerns that were raised about terminology.  First, we agree with the concern about the use of the term “reasoning,” as our tasks did not involve any explicit sequential processing.  Accordingly, we have decided to use the term “inference module” instead of “reasoning module”. Regarding “attention,” while it is true that, in the model presented, the selection of the grid cells is not context-dependent (e.g., specific to the input), as might be expected for an attentional mechanism, nevertheless it is a selective (“filtering”) mechanism that limits the scope of sensory information subjected to further processing, and we envision that in future work this could be made to be context-specific (e.g., selecting different grid cells for different inputs). For these reasons, our preference is to continue to use the term “attention”, explicitly acknowledging that while, in its present form, the mechanism is context-insensitive, making it context-sensitive would be a valuable direction for future research.  That said, if the reviewer continues to feel strongly about this point, we would be willing to consider alternative terms, such as “filter” or “bias.”
>
> Finally, we agree it is important to clarify how we use the term “extrapolation”, and in response to reviewers’ helpful comments, we are currently modifying the definition of “extrapolation” we assume in the paper. Briefly, it has been argued ([1]) that in high dimensions virtually all generalizations can be considered extrapolation (i.e., most points not in the training set can be described as being outside the convex hull of the training set). However, we would like to point out that this argument rests on a specific geometric definition of the convex hull (viz., it is formed as the simplicial complex of the training points) and how this relates to the assumption about the generative model underlying the data. If the data are assumed to come from a simplicial complex of a similar form, then the point stands.  However, if they are assumed to come from some smoother function (as is likely to be the case for many if not most forms of naturalistic data), then the convex hull is too restrictive a condition, and a more appropriate “boundary” would be a smoothed function of the training data, which would encompass many more points within its bounds that were not in the training set.  In such cases, a simple, but conservative, assumption is to consider the boundary as a hypersphere, the center of which is the centroid of the training data, and the radius of which is the maximum of the norms of the vectors from the centroid to each training data point.  We believe this captures the sense in which “extrapolation” is usually meant (that is, falling “outside the range” of the training data). The training and test data used in our tasks satisfy this definition of extrapolation, and we are currently modifying the paper to clearly reflect this definition of extrapolation.  However, again, if the reviewer feels strongly that the term “extrapolation” is misguided, even under this revised conception, we would welcome other terminological suggestions, and in any event, will try to convey this point more clearly in our revision.
>
> [1] - Balestriero, R., Pesenti, J. and LeCun, Y., 2021. Learning in high dimension always amounts to extrapolation. arXiv preprint arXiv:2110.09485.

---

> > ### Comment · Reviewer_6GAB · 2023-02-19
> > **Terminology**
> >
> > I do feel strongly that reasoning, attention and extrapolation are words that have a clear meaning and that their use in the paper does not match these meanings. In particular, extrapolation has a formal meaning, as in the cited paper [1] (but also much earlier). I agree that people sometimes use it colloquially in a more fuzzy way, but in papers in applied mathematical sciences like computer science and machine learning, we should really stick to the clear and established scientific meaning. The case of attention is more central to this paper. It really is a misnomer as used. There is no sense in which attention would be context-insensitive or static. There are other words for when a selection or modulation of weights is context-insensitive and static. Machine learners use words like weight pruning for example, or feature selection. Please use the words that correspond to your usage and that are established in our community. It's fine to talk about attention in future work to extend the current static selection to a context-conditioned one, though.
> >
> > The hard problem that remains with this paper is whether it is appropriate for TMLR's audience even though it only addresses 2-dimensional input with no clear hope or path in my opinion that this approach will generalize to higher-dimensional inputs.

---

> > > ### Author Response · Authors · 2023-02-21
> > > **Followup regarding terminology**
> > >
> > > We thank the reviewer for their helpful suggestions and comments. We have decided to change the terminologies (https://openreview.net/forum?id=6IeUepLl21&noteId=N1pJVQbLwj) in the updated version of our manuscript. We hope it addresses the reviewer's concerns.

---

> > > > ### Comment · Reviewer_6GAB · 2023-02-22
> > > > **terminology**
> > > >
> > > > Yes, it addresses my concerns about terminology. Thank you.

---

> ### Author Response · Authors · 2023-02-21
> **Followup regarding appropriateness for TMLR audience**
>
> We thank the reviewer for their helpful suggestions and comments. We have attempted to clarify the contribution of the paper, potential ways of extending it to more practical applications, and value for the TMLR audience (https://openreview.net/forum?id=6IeUepLl21&noteId=sxJbyB8lLP). We hope it addresses the reviewer's concerns.

---

> ### Author Response · Authors · 2023-03-01
> **Response to reviewer 6GAB**
>
> We thank the reviewer for their insightful feedback and suggestions. We have uploaded a revised version of the manuscript. Below we include a detailed response to the reviewers' comments.
>
> -*clarity needs improvements in several places, including regarding the reason why the DPP formula would be helpful.*
>
> We thank the reviewer for this suggestion. The second component of our proposed framework involves selecting grid embeddings that promote out-of-distribution generalization. Specifically, the algorithm is designed to select, which we identify as the grid embeddings that exhibit the greatest within frequency variance but are least redundant (that is, that are least also pairwise uncorrelated) over the training data. This is achieved by maximizing the determinant of the covariance matrix over the within frequency grid embeddings of the training data.  However finding the absolute maximum of this optimization problem is NP-hard. Hence we use a continuous approximation of this discrete optimization problem as proposed by Gilenwater et al., 2012, which is given in equation 4. We use Theorem 2.1 to construct $\mathcal{L}_{DPP}$ which is basically obtained by applying log on both sides of the theorem, which is the exactly the function we want to maximize, $f(x) = \log (\det(V_x))$, where $V_x$ is the covariance matrix of the grid embeddings of the grid cells indexed by $x$, here in our case within a particular frequency. We have clarified this in section 2.2.2.
>
> Reference:
>
> Gillenwater, J., Kulesza, A. and Taskar, B., 2012. Near-optimal map inference for determinantal point process
>
> -*The method should be validated on tasks that exist in the ML literature (where others have done their best to get a good result) and should not be limited to so low-dimensional data, for relevance to the TMLR audience.*
>
> We appreciate the concerns raised by the reviewer, which have motivated us to more clearly articulate the purpose of the paper.  This paper focuses on the challenge of understanding how the capacity for generalization arises in natural neural systems, which may inspire future work addressing how this can be applied in artificial neural networks.  Specifically  the purpose of the paper is to demonstrate how particular properties of processing observed in the brain – in particular, the use of grid cell embeddings –  can be used to achieve strong out-of-distribution generalization, when combined with a generic selective filter over those embeddings. For proof of concept, we focused on tasks involving two dimensional stimuli, which have similarly been the focus of work in neuroscience (i.e., spatial location), and therefore serve as a reasonable starting point for examining the properties of interest in simulated networks. Based on our findings, we believe that the mechanisms  we have identified should apply readily to higher dimensional stimuli, and therefore have significant potential for broader future practical applications. However,  in order to extend this  approach to more practical domains (e.g. involving end-to-end training), methods would need to be developed to extract grid-like codes from such high-dimensional real-world data.  This remains a current challenge, although several recent results (Wei et al., 2015; Cueva & Wei, 2018; Banino et al., 2018;Whittington et al., 2020)
> suggest that this should be possible. We have revised the paper so as to clarify the framing of our contribution and potential for more practical applications. The relevant sentences are:
>
> ### From the Abstract
>
> “Here, we identify how certain properties of processing in the brain can be used to achieve strong OOD generalization and offer a two-part algorithm that improves OOD generalization performance of artificial neural networks on multiple tasks widely used in neuroscience to demonstrate proof of concept.”
>
> ### From the Introduction
>
> “ Here, we consider two cognitive problems that are widely used in neuroscience and often require a capacity for OOD generalization: a) analogy and b) arithmetic. What enables the human brain to successfully generalize on these tasks, and how might we better realize that ability in deep learning systems?”
>
> “Taken together, the representational and selection mechanisms outlined above define a two-component framework for promoting OOD generalization, by minimizing task-specific error subject to: i) embeddings that encode relational structure among the data (grid cells), and ii) selection of those embeddings that maximize the "volume" of the representational space that is covered, while minimizing redundancy (DPP-S). Below, we demonstrate proof of concept by showing that these mechanisms allow artificial neural networks to learn representations that support OOD generalization on two cognitive tasks which have similarly been the focus in neuroscience and therefore serve as a reasonable starting point for examining the properties of interest in these networks.”

---

> > ### Author Response · Authors · 2023-03-01
> > **Response to reviewer 6GAB (Part 2)**
> >
> > ### From the Discussion and future directions
> >
> > “We have identified how particular properties of processing observed in the brain can be used to achieve strong OOD generalization, and introduced a two-component algorithm to promote OOD generalization in deep neural networks.”
> >
> > “For proof of concept, we started with two cognitive tasks widely used in neuroscience (analogy and arithmetic), and showed that the combination of grid code and DPP-S promotes OOD generalization across both translation and scale when incorporated into common architectures (LSTM and transformer).”
> >
> > “The current approach may be seen to be limited by the fact that we derive the grid codes from known properties of neural systems, rather than obtaining these codes directly from real-world data. Here, we are encouraged by the body of work providing evidence for grid-like codes in the hidden layers of neural networks in a variety of task contexts and architectures (Wei et al., 2015; Cueva & Wei, 2018; Banino et al., 2018;Whittington et al., 2020). This suggests reason for optimism that DPP-S may promote strong generalization in cases where grid-like codes naturally emerge: for example, navigation tasks (Banino et al., 2018) and reasoning by transitive inference (Whittington et al., 2020). Integrating our approach with structured representations acquired from high-dimensional, naturalistic datasets remains a critical next step which would have significant potential for broader future practical applications. So too does application to more complex transformations beyond translation and scale, such as rotation.”
> >
> > References:
> >
> >
> >  Wei, X.X., Prentice, J. and Balasubramanian, V., 2015. A principle of economy predicts the functional architecture of grid cells. Elife, 4, p.e08362.
> >
> > Cueva, C.J. and Wei, X.X., 2018. Emergence of grid-like representations by training recurrent neural networks to perform spatial localization. arXiv preprint arXiv:1803.07770.
> >
> > Banino, A., Barry, C., Uria, B., Blundell, C., Lillicrap, T., Mirowski, P., Pritzel, A., Chadwick, M.J., Degris, T., Modayil, J. and Wayne, G., 2018. Vector-based navigation using grid-like representations in artificial agents. Nature, 557(7705), pp.429-433.
> >
> > Whittington, J.C., Muller, T.H., Mark, S., Chen, G., Barry, C., Burgess, N. and Behrens, T.E., 2020. The Tolman-Eichenbaum machine: unifying space and relational memory through generalization in the hippocampal formation. Cell, 183(5), pp.1249-1263.
> >
> >
> >
> >
> > Given this focus, we believe that our paper falls within TMLR’s scope, as articulated at  https://www.jmlr.org/tmlr/editorial-policies.html and, accordingly, we anticipate that the findings will be of interest to TMLR’s audience. Specifically, our paper addresses:
> >
> > a) experimental and/or theoretical studies yielding new insight into the design and behavior of learning in intelligent systems;
> >
> > b) computational models of natural learning systems at the behavioral or neural level;
> >
> > -*2nd par. of intro, points (a) and (b). What is meant by "structured nature of representations" and "representational volume" is not yet clear at this point in the paper.*
> >
> > We agree that these weren’t clear in our initial submission. We have attempted to clarify in the revised manuscript. The 2nd par. of the intro now reads:
> >
> > To address the problem, we focus on two properties that we hypothesize are useful for OOD generalization in biological systems: a) the *representations* of relational structure, in which relations are preserved across transformations like translation and scaling (such as observed for grid cells in mammalian medial entorhinal cortex (Hafting et al., 2005)); and b)  a *selectional objective* to select representations that have maximum variance and minimum correlation among them, over the training data. The net effect of these two properties is to normalize the representations of training and testing data in a way that preserves their relational structure, and allows the network to learn that structure in a form that can be applied well beyond the domain over which it was trained.
> >
> > Reference:
> >
> > Hafting, T., Fyhn, M., Molden, S., Moser, M.B. and Moser, E.I., 2005. Microstructure of a spatial map in the entorhinal cortex. Nature, 436(7052), pp.801-806.
> >
> > -*p. 2, 1st par., "invariant to translation within frequency and invariant to scale across frequencies" only becomes clear later, but the reader hasn't read those things yet.*
> >
> > We have now clarified these statements in the 2nd page, 1st par. The sentence now reads- “Of interest here, the periodic response function displayed by grid cells belonging to a particular frequency is invariant to translation by its period, and increasing the scale of a higher frequency response gives a lower frequency response and vice versa, making it invariant to scale across frequencies.”

---

> > > ### Author Response · Authors · 2023-03-01
> > > **Response to reviewer 6GAB (Part 3)**
> > >
> > > -*p. 2, 2nd par., "within context variance ("relevance")" is not clear at this point*
> > >
> > > We agree that this wasn't clear in our initial submission. We have rephrased to simply state “greatest variance”.
> > >
> > > -*p. 5, 1st par., "conceptual states" is too vague, and seems to go against common sense that concepts live in a high-dimensional space, not in a 2-D space (except for the very specific concept of spatial location!)*
> > >
> > > We agree this was a bit vague. We have rephrased “conceptual states” to “conceptual knowledge in two continuous dimensions”. We would like to clarify that it's meaningful to consider a very low-dimensional subset of a high dimensional conceptual space that is behaviorally relevant.
> > >
> > >
> > > -*p. 5, Eq. 2: F is not defined/or explained (what are its dimensions?), nor is A, nor is A_offset.*
> > >
> > >
> > > We have now explained the following terms in the last paragraph of section 2.2.1 of the revised manuscript. The relevant sentences are:
> > >
> > > “The spatial frequencies of grids ($F$) begin at a value of $0.0028*2\pi$.”
> > >
> > > “ Here, we choose $N_f=9$ (dimension of $F$) as the number of frequencies.”
> > >
> > >  “$A$ refers to a particular location in a two dimensional space, and 100 offsets ($A_{offset}$) are used for each frequency to evenly cover a space of $1000\times1000$ locations using 900 grid cells. These offsets represent different phases within each frequency and since there are 100 of them, $N_p = 100$”
> > >
> > > -*section 2.2.2 (DPP-A): the connection between the literature on DPP and the proposed architecture is too vague. Why is a such a selection of grid points expected to be useful? the questions below about algorithm 1 should be answered here. algorithm 1: why is this choice of frequency f a good thing? why do we even want to select a frequency? (this should be explained in the text)*
> > >
> > > We have attempted to more strongly connect our proposed architecture with the literature on DPP and provided an intuition of why a selection of grid points based on a particular choice of frequency is expected to be useful in the first two added paragraphs of section 2.2.2. They read:
> > >
> > > “We hypothesize that the use of a relational encoding metric (i.e., grid code) is extremely useful, but not sufficient for a system to achieve strong generalization, which requires selecting particular aspects of the encoding that can capture the same relational structure across the training and test distributions. Toward this end, we propose a selectional objective that uses the statistics of the training data to select the grid embeddings that can induce the inference module to achieve strong generalization. Our objective, which we describe in detail below, seeks to identify those grid embeddings that exhibit the greatest variance but are least redundant (pairwise uncorrelated over the training data). Formally, this is captured by maximizing the determinant of the covariance matrix of the grid embeddings computed over the training data (Kulesza & Taskar, 2012). Although in machine learning, DPPs have been particularly influential in work on recommender systems (Chen et al., 2018, summarization (Gong et al., 2014; Perez-Beltrachini & Lapata, 2021),neural network pruning (Mariet & Sra, 2015), here, we propose to use maximization of the determinant of the covariance matrix as a filtering mechanism to limit the influence of grid embeddings with low-variance (which are less relevant) or with high-similarity to other grid embeddings (which are redundant).”
> > >
> > > "For the specific tasks that we study here, we have assumed the grid embeddings to be pre-learned to represent the entire space of possible test data points, and we are simply focused on the problem of how to determine which of these are most useful in enabling generalization for a task-optimized network trained on a small fraction of that space (Figure 2). That is, we look for a way to select a subset of grid-cells frequencies whose embeddings capture recurring task-relevant relational structure. We find that grid embeddings corresponding to the higher spatial frequency grid cells exhibit more variance on the training data than the low frequency embeddings. In any training set, low spatial frequency embeddings may carry information about the stimuli that can be used to help minimize task-error, but, critically, the higher frequency embeddings due to their higher variance tend to preferentially capture the same relational structure across different regions which is necessary for OOD generalization. Accordingly, we find that filtering for the determinant maximizing grid cell embeddings tends to select those that encode higher frequencies."

---

> > > > ### Author Response · Authors · 2023-03-01
> > > > **Response to reviewer 6GAB (Part 4)**
> > > >
> > > >   References:
> > > >
> > > > Kulesza, A. and Taskar, B., 2012. Determinantal point processes for machine learning. Foundations and Trends® in Machine Learning, 5(2–3), pp.123-286.
> > > >
> > > > Chen, L., Zhang, G. and Zhou, E., 2018. Fast greedy map inference for determinantal point process to improve recommendation diversity. Advances in Neural Information Processing Systems, 31.
> > > >
> > > > Gong, B., Chao, W.L., Grauman, K. and Sha, F., 2014. Diverse sequential subset selection for supervised video summarization. Advances in neural information processing systems, 27.
> > > >
> > > > Perez-Beltrachini, L. and Lapata, M., 2021. Multi-document summarization with determinantal point process attention. Journal of Artificial Intelligence Research, 71, pp.371-399.
> > > >
> > > > Mariet, Z. and Sra, S., 2015. Diversity networks: Neural network compression using determinantal point processes. arXiv preprint arXiv:1511.05077.

---

### Review · Reviewer_3smA · 2023-02-14

**Summary Of Contributions:**

This paper introduces a new biologically-inspired encoding scheme based on grid cells in the brain along with an attention mechanism over the encoding to learn a relational embedding for inputs that promotes extrapolation in downstream tasks. The attention mechanism is also biologically inspired and effectively learns a Determinantal Point Process over grid cells to maximize relevance and diversity measured by a smooth approximation of the determinant of the covariance matrix.  Experiments on learning translation and scale invariance and arithmetic demonstrate the promise of this grid embedding in enabling extrapolation.

**Audience:**

Yes

**Broader Impact Concerns:**

None.

**Claims And Evidence:**

No

**Requested Changes:**

- Critical: Please provide more details about how the grid embeddings are formed.  I am still very confused how you go from a point in 2D space to the grid embeddings.  How does 900 frequency and phase settings get mapped to a grid with 1000x1000 locations?
- Critical: Please explain why the frequency passed to the reasoner is $f_{max_{DDP}}$ in Algorithm 1 (i.e. $\hat{y}=R(x_f=f_{max_{DDP})$).
- Critical: Please provide more details about how the loss is computed and how the foils are used.
- Critical: Please include ablations studies that show performance when increasing the number of frequencies considered and also perhaps more complicated grid embedding architecture to see if multiplication can be solved more effectively.
- Qn: why is an one-hot vector required to differentiate between addition and multiplication for the arithmetic task?  Is the reasoner learning both addition and multiplication simultaneously or do you train separate networks to learn addition and multiplication?


**Strengths And Weaknesses:**

Strengths:
- This paper presents an interesting view of extrapolation and relationship learning motivated from discoveries in neuroscience.

Weaknesses:
- The methodology section is very confusing and, despite trying my best, I could not understand how the grid embeddings are computed for an input data point.
- The approach is very restrictive in its current form and can only be applied points arising in bounded grids.  It is unclear how useful this will be in general.  How would the method be applied to more practical problems and data types (e.g. higher dimensional vectors)?
- Results are for very toy problems: 1. predicting a point D in 2D space that would result in completing a vector CD of the same length as a known vector AB and 2. performing elementwise addition and multiplication on 2D vectors.
- The writing can be improved overall for clarity.  This is mainly a point of feedback for the introduction where there are many instances where the authors would explain something in a way that is not informative without knowledge of prior work or use terms that are not explained until later.
- The method is not able to fully extrapolate on multiplication.

---

> ### Author Response · Authors · 2023-02-19
> **Clarification for Reviewer 3smA**
>
> We thank the reviewer for their helpful suggestions and comments. We are in the process of addressing the requested changes, and will get back with a more detailed response soon. Toward that end, we would like to ask for clarification concerning the more complicated forms of grid embedding architecture that the reviewer had in mind.

---

> ### Author Response · Authors · 2023-03-01
> **Response to Reviewer 3smA**
>
> We thank the reviewer for their insightful feedback and suggestions. We have uploaded a revised version of the manuscript. Below we include a detailed response to the reviewers' comments.
>
> -*The approach is very restrictive in its current form and can only be applied points arising in bounded grids. It is unclear how useful this will be in general. How would the method be applied to more practical problems and data types (e.g. higher dimensional vectors)?*
>
> We appreciate the concerns raised by the reviewer, which have motivated us to more clearly articulate the purpose of the paper.  This paper focuses on the challenge of understanding how the capacity for generalization arises in natural neural systems, which may inspire future work addressing how this can be applied in artificial neural networks.  Specifically, the purpose of the paper is to demonstrate how particular properties of processing observed in the brain – in particular, the use of grid cell embeddings –  can be used to achieve strong out-of-distribution generalization, when combined with a generic selective filter over those embeddings. For proof of concept, we focused on tasks involving two dimensional stimuli , which have similarly been the focus of work in neuroscience (i.e., spatial location), and therefore serve as a reasonable starting point for examining the properties of interest in simulated networks. Based on our findings, we believe that the mechanisms  we have identified should apply readily to higher dimensional stimuli, and therefore have significant potential for broader future practical applications. However,  in order to extend this  approach to more practical domains (e.g. involving end-to-end training), methods would need to be developed to extract grid-like codes from such high-dimensional real-world data.  This remains a current challenge, although several recent results (Wei et al., 2015; Cueva & Wei, 2018; Banino et al., 2018;Whittington et al., 2020)
>  suggest that this should be possible. We have revised the paper so as to clarify the framing of our contribution and potential for more practical applications. The relevant sentences are:
>
> ### From the Abstract
>
> “Here, we identify how certain properties of processing in the brain can be used to achieve strong OOD generalization and offer a two-part algorithm that improves OOD generalization performance of artificial neural networks on multiple tasks widely used in neuroscience to demonstrate proof of concept.”
>
> ### From the Introduction
>
> “ Here, we consider two cognitive problems that are widely used in neuroscience and often require a capacity for OOD generalization: a) analogy and b) arithmetic. What enables the human brain to successfully generalize on these tasks, and how might we better realize that ability in deep learning systems?”
>
> “Taken together, the representational and selection mechanisms outlined above define a two-component framework for promoting OOD generalization, by minimizing task-specific error subject to: i) embeddings that encode relational structure among the data (grid cells), and ii) selection of those embeddings that maximize the "volume" of the representational space that is covered, while minimizing redundancy (DPP-S). Below, we demonstrate proof of concept by showing that these mechanisms allow artificial neural networks to learn representations that support OOD generalization on two cognitive tasks which have similarly been the focus in neuroscience and therefore serve as a reasonable starting point for examining the properties of interest in these networks.”

---

> > ### Author Response · Authors · 2023-03-01
> > **Response to Reviewer 3smA (Part 2)**
> >
> > ### From the Discussion and future directions
> >
> > “We have identified how particular properties of processing observed in the brain can be used to achieve strong OOD generalization, and introduced a two-component algorithm to promote OOD generalization in deep neural networks.”
> >
> > “For proof of concept, we started with two cognitive tasks widely used in neuroscience (analogy and arithmetic), and showed that the combination of grid code and DPP-S promotes OOD generalization across both translation and scale when incorporated into common architectures (LSTM and transformer).”
> >
> > “The current approach may be seen to be limited by the fact that we derive the grid codes from known properties of neural systems, rather than obtaining these codes directly from real-world data. Here, we are encouraged by the body of work providing evidence for grid-like codes in the hidden layers of neural networks in a variety of task contexts and architectures (Wei et al., 2015; Cueva & Wei, 2018; Banino et al., 2018;Whittington et al., 2020). This suggests reason for optimism that DPP-S may promote strong generalization in cases where grid-like codes naturally emerge: for example, navigation tasks (Banino et al., 2018) and reasoning by transitive inference (Whittington et al., 2020). Integrating our approach with structured representations acquired from high-dimensional, naturalistic datasets remains a critical next step which would have significant potential for broader future practical applications. So too does application to more complex transformations beyond translation and scale, such as rotation.”
> >
> >
> > References are:
> >
> >
> >  Wei, X.X., Prentice, J. and Balasubramanian, V., 2015. A principle of economy predicts the functional architecture of grid cells. Elife, 4, p.e08362.
> >
> > Cueva, C.J. and Wei, X.X., 2018. Emergence of grid-like representations by training recurrent neural networks to perform spatial localization. arXiv preprint arXiv:1803.07770.
> >
> > Banino, A., Barry, C., Uria, B., Blundell, C., Lillicrap, T., Mirowski, P., Pritzel, A., Chadwick, M.J., Degris, T., Modayil, J. and Wayne, G., 2018. Vector-based navigation using grid-like representations in artificial agents. Nature, 557(7705), pp.429-433.
> >
> > Whittington, J.C., Muller, T.H., Mark, S., Chen, G., Barry, C., Burgess, N. and Behrens, T.E., 2020. The Tolman-Eichenbaum machine: unifying space and relational memory through generalization in the hippocampal formation. Cell, 183(5), pp.1249-1263.
> >
> >
> > -*The writing can be improved overall for clarity. This is mainly a point of feedback for the introduction where there are many instances where the authors would explain something in a way that is not informative without knowledge of prior work or use terms that are not explained until later.*
> >
> > We apologize for the lack of clarity in writing and appreciate this point. We have rewritten the introduction to reflect this. The clarified sentences are:
> >
> > “To address the problem, we focus on two properties that we hypothesize are useful for OOD generalization in biological systems: a) the *representations* of relational structure, in which relations are preserved across transformations like translation and scaling (such as observed for grid cells in mammalian medial entorhinal cortex (Hafting et al., 2005)); and b) a *selectional objective* to select representations that have maximum variance and minimum correlation among them,  over the training data. The net effect of these two properties is to normalize the representations of training and testing data in a way that preserves their relational structure, and allows the network to learn that structure in a form that can be applied well beyond the domain over which it was trained. “
> >
> > “Here, we test whether the same capabilities can be achieved using a well-established, biologically plausible embedding scheme -- grid codes -- and an adaptive form of normalization that is based strictly on the statistics of the training data in the embedding space. We show that when deep neural networks
> > are presented with data that exhibits such relational structure, grid code embeddings coupled with an error-minimizing/selectional objective promotes strong OOD generalization. We unpack each of these theoretical components in turn before describing the tasks, modeling architectures, and results.”
> >
> > “**Representations of Relational Structure.** The first component of the proposed framework relies on the idea that a key element underlying human-like OOD generalization is the use of low-dimensional representations that emphasize the relational structure between data points.”
> >
> > “Of interest here, the periodic response function displayed by grid cells belonging to a particular frequency is invariant to translation by its period, and increasing the scale of a higher frequency response gives a lower frequency response and vice versa, making invariant to scale across frequencies”

---

> > > ### Author Response · Authors · 2023-03-01
> > > **Response to Reviewer 3smA (Part 3)**
> > >
> > > “Despite the use of a relational encoding metric (i.e., grid code), generalization may also require identifying which aspects of this encoding that could potentially be shared across training and test distributions. Here, we implement this by identifying, and restricting further processing to those grid embeddings
> > > that exhibit the greatest variance, but are least redundant (that is, pairwise uncorrelated) over the training data.”
> > >
> > >
> > > “DPPs have since been adopted in machine learning for applications in which diversity in a subset of selected items is desirable, such as recommender systems (Kulezka & Taskar, 2012). Recent work in computational cognitive science has shown DPPs naturally capture inductive biases in human inference, such as some word-learning and reasoning tasks (e.g., one noun should only refer to one object) while also serving as an efficient memory code (Frankland & Cohen, 2020). In that context, the learner is biased to find a set of possible word-meaning pairs whose representations exhibit the greatest variance and lowest covariance on a task-relevant dataset. DPPs also provide a formal objective for the type of orthogonal coding that has been proposed to be characteristic of representations in mammalian hippocampus, and integral for episodic memory (McClelland et al., 1995). Thus, using the DPP objective to govern selection of grid code representations, known to be implemented in the entorhinal cortex (one synapse upstream of the hippocampus), aligns with the function and organization of cognitive and neural systems underlying the capability for abstraction.”
> > >
> > > “Below, we demonstrate proof of concept by showing that these mechanisms allow artificial neural networks to learn representations that support OOD generalization on two cognitive tasks which have similarly been the focus in neuroscience and therefore serve as a reasonable starting point for examining the properties of interest in these networks.”
> > >
> > > References:
> > >
> > > Hafting, T., Fyhn, M., Molden, S., Moser, M.B. and Moser, E.I., 2005. Microstructure of a spatial map in the entorhinal cortex. Nature, 436(7052), pp.801-806.
> > >
> > >  Kulesza, A. and Taskar, B., 2012. Determinantal point processes for machine learning. Foundations and Trends® in Machine Learning, 5(2–3), pp.123-286.
> > >
> > > Frankland, S. and Cohen, J., 2020. Determinantal Point Processes for Memory and Structured Inference. In CogSci.
> > >
> > > McClelland, J.L., McNaughton, B.L. and O'Reilly, R.C., 1995. Why there are complementary learning systems in the hippocampus and neocortex: insights from the successes and failures of connectionist models of learning and memory. Psychological review, 102(3), p.419.
> > >
> > >
> > > -*Critical: Please provide more details about how the grid embeddings are formed. I am still very confused how you go from a point in 2D space to the grid embeddings. How does 900 frequency and phase settings get mapped to a grid with 1000x1000 locations?*
> > >
> > > Again, we apologize for the lack of clarity. We created 900 grid cells firing rate maps over a space of $1000 \times1000$ locations using the publicly available codebase of Bicanksi & Burgess (2019), which are superimposed cosine waves spanning 9 distinct frequencies and 100 phases created by different offsets. Each 2d point in a grid with $1000\times1000$ locations was represented by the firing rate of 900 grid cells, which constituted the grid embedding for that 2d point. We have added more details about this in section 2.2.1.
> > >
> > > Reference:
> > >
> > > Bicanski, A. and Burgess, N., 2019. A computational model of visual recognition memory via grid cells. Current Biology, 29(6), pp.979-990.
> > >
> > >
> > > -*Critical: Please explain why the frequency passed to the reasoner is $f_{max_{DPP}}$ in Algorithm 1 (i.e. $\hat{y}=R(x_f=f_{max_{DDP})$).*
> > >
> > > This implements the second component of our proposed framework, which involves selecting grid embeddings that promote out-of-distribution generalization.  Specifically, the algorithm is designed to select the grid embeddings that exhibit the greatest within frequency variance but are least redundant (that is, that are least also pairwise uncorrelated) over the training data. This is achieved  by maximizing the determinant of the covariance matrix over the within frequency grid embeddings of the training data. $f_{max_{DPP}}$ is the frequency of the grid embeddings that had the maximum determinant as computed by  ${\hat{F}}_{f}$
> > >
> > > for each frequency $f \in [1, N_f]$, after maximizing the objective function in Equation 6. Hence we pass the grid embeddings corresponding to frequency  $f_{max_{DPP}}$ to the reasoner, which we find are best at capturing the relational structure across the training and testing data, thereby promoting out-of-distribution generalization. We have clarified this in the paper in section 2.2.2.

---

> > > > ### Author Response · Authors · 2023-03-01
> > > > **Response to Reviewer 3smA (Part 4)**
> > > >
> > > > -*Critical: Please provide more details about how the loss is computed and how the foils are used.*
> > > >
> > > >  We have added more details about $\mathcal{L}_{task}$ and how foils are used in section 2.2.3. The relevant sentences with the additional details are:
> > > >
> > > > “For each analogy, the inference module was presented with multiple candidate
> > > > problems, each consisting of three stimuli, $A,B,C$, and a set containing $D$ (the correct completion) and six foil completions. For each instance of the arithmetic task, it was presented with two  stimuli, $A,B$, and a set containing $C$ (the correct completion) and six foil completions. A linear output layer was used to generate a score for the candidate completions for each problem.”
> > > >
> > > > “The seven scores (one for the correct completion and for six foil completions) were normalized using a softmax function, such that higher score would correspond to higher probablity and vice versa and the probabilities sum to 1. The inference module was trained using the task specific cross entropy loss ($\mathcal{L}_{task}= \operatorname{cross-entropy}$) between the softmax-normalized scores  and the index for the correct completion (target).”
> > > >
> > > > -*Critical: Please include ablations studies that show performance when increasing the number of frequencies considered and also perhaps more complicated grid embedding architecture to see if multiplication can be solved more effectively.*
> > > >
> > > > We thank the reviewer for suggesting the additional ablation studies. We have added the ablation study when increasing the number of grid cell frequencies $N_f$ in the Appendix, Figure 13 which shows that the average test accuracy on multiplication improves by around 5% when increasing $N_f$ from 9 to 11/12, but still it is much lower than the near perfect accuracy on addition. These results demonstrate that just increasing number of grid cell frequencies isn’t able to capture the distributional characteristics that is required to solve multiplication more effectively. We would like to ask for clarification concerning the more complicated forms of grid architecture that the reviewer had in mind.
> > > >
> > > >
> > > > -*Qn: why is an one-hot vector required to differentiate between addition and multiplication for the arithmetic task? Is the reasoner learning both addition and multiplication simultaneously or do you train separate networks to learn addition and multiplication?*
> > > >
> > > >  The one-hot “task” input in required because the reasoner is indeed trained simultaneously to do both addition and multiplication.  Thus, the one-hot vector is needed to distinguish which task must be performed.

---

> > ### Comment · Reviewer_6GAB · 2023-03-01
> > **Belief in applicability of shown results to higher dimensions**
> >
> > You wrote in your response "Based on our findings, we believe that the mechanisms we have identified should apply readily to higher dimensional stimuli". However, I (and apparently the other reviewers) do not see any argument or evidence to also believe that. Without sufficient reasons to believe this, it is not clear if this work has any bearing on machine learning, or even in the ability of the brain to handle higher-dimensional stimuli.

---

> > > ### Author Response · Authors · 2023-03-03
> > > **Extension to higher-dimensional inputs**
> > >
> > > We thank the reviewer for their continued engagement on this topic. We respectfully disagree with the reviewer’s claim that we have not provided any argument to support the plausibility of extending our work to higher-dimensional inputs. Our argument is as follows:
> > >
> > > 1) There is a wealth of evidence that the brain uses low-dimensional grid-like codes to represent relational structure, both in the spatial domain [1], but also crucially in non-spatial domains, such as for visual [2] or auditory [3] input. In these latter cases, though the inputs themselves are high-dimensional, they can and are represented by the brain in a low-dimensional latent space, sometimes consisting of two [2] or even just one dimension [3]. We address this point in the section of the paper entitled 'Representations of Relational Structure', and also in the first paragraph of section 2.2.1
> > >
> > > 2) Recent work has demonstrated that it is possible to extract these kinds of low-dimensional codes directly from high-dimensional inputs [4,5,6]. This work provides a plausible computational account for how the brain might extract such codes from naturalistic inputs. This point is addressed in the final paragraph of the discussion.
> > >
> > > We agree that it is very important to try and integrate these approaches with the approach that we propose in the present work, and we intend to do so in future work. But our aim in the present work was to address the distinct question of how these codes might contribute to out-of-distribution generalization. Our results show that these kinds of representations are necessary, *but not sufficient*, to achieve strong out-of-distribution generalization. That is, even given access to idealized grid codes, we found that models were not able to extrapolate effectively without the additional incorporation of our proposed DPP selection method. Therefore, our contribution is distinct from, but compatible with, other work that has addressed the question of how to extract these codes from high-dimensional inputs. We do not see any particular reason for pessimism about the prospects for integrating these two lines of work.
> > >
> > > We also disagree with the reviewer’s claim that our work does not fall within the scope of TMLR. Our understanding is that TMLR is explicitly conceived as a venue not only for work that addresses practical machine learning applications, but also for work that addresses 'computational models of *natural learning systems* at the behavioral or neural level' and that contributes to 'the understanding of the computational and mathematical principles that enable intelligence through learning, *be it in brains or in machines*.' We believe that our work clearly meets these criteria, and will be of interest both to the neuroscience and machine learning communities.
> > >
> > > [1] Torkel Hafting, Marianne Fyhn, Sturla Molden, May-Britt Moser, and Edvard I Moser. Microstructure of a spatial map in the entorhinal cortex. Nature, 436(7052):801–806, 2005.
> > >
> > > [2] Alexandra O Constantinescu, Jill X O’Reilly, and Timothy EJ Behrens. Organizing conceptual knowledge in humans with a gridlike code. Science, 352(6292):1464–1468, 2016.
> > >
> > > [3] Dmitriy Aronov, Rhino Nevers, and David W Tank. Mapping of a non-spatial dimension by the hippocampal–entorhinal circuit. Nature, 543(7647):719–722, 2017.
> > >
> > > [4] Christopher J Cueva and Xue-Xin Wei. Emergence of grid-like representations by training recurrent neural networks to perform spatial localization. arXiv preprint arXiv:1803.07770, 2018.
> > >
> > > [5] Andrea Banino, Caswell Barry, Benigno Uria, Charles Blundell, Timothy Lillicrap, Piotr Mirowski, Alexander Pritzel, Martin J Chadwick, Thomas Degris, Joseph Modayil, et al. Vector-based navigation using grid-like representations in artificial agents. Nature, 557(7705):429–433, 2018.
> > >
> > > [6] James CR Whittington, Timothy H Muller, Shirley Mark, Guifen Chen, Caswell Barry, Neil Burgess, and Timothy EJ Behrens. The tolman-eichenbaum machine: Unifying space and relational memory through generalization in the hippocampal formation. Cell, 183(5):1249–1263, 2020.

---

### Author Response · Authors · 2023-02-21
**Clarification of our contribution, extension to more practical applications, and value for TMLR audience**

We would like to clarify that the purpose of the paper is primarily conceptual, demonstrating how certain principles associated with neural processing can, when appropriately combined, achieve strong out-of-distribution generalization – principles that, with further examination and development, may also prove useful in the design of more flexible artificial systems. For proof of concept, we used two-dimensional tasks that have been extensively in the neuroscientific work on which we draw, and that demonstrate how generalization in the human brain might work. We agree that our findings do not make an immediate practical contribution to the design of artificial systems, which are often concerned with higher dimensional inputs, but it seems reasonable to believe that the principles investigated in this work should extend to such inputs. The challenge is in developing methods that can extract grid-like codes from high-dimensional real-world data. This is a challenge that has not yet been fully met in computational neuroscience but a number of recent results ([1,2,3]) suggest that this should be possible. Our work would provide an immediate foundation on which to extend such results to explore mechanisms for extrapolation in such higher dimensional spaces.

Finally, we would believe that our paper falls within TMLR’s scope, as articulated at  https://www.jmlr.org/tmlr/editorial-policies.html and, accordingly, we anticipate that the findings will be of interest to TMLR’s audience. Specifically, our paper addresses:

a) experimental and/or theoretical studies yielding new insight into the design and behavior of learning in intelligent systems

b) computational models of natural learning systems at the behavioral or neural level



[1] - Cueva, C.J. and Wei, X.X., 2018. Emergence of grid-like representations by training recurrent neural networks to perform spatial localization. arXiv preprint arXiv:1803.07770.

[2] - Banino, A., Barry, C., Uria, B., Blundell, C., Lillicrap, T., Mirowski, P., Pritzel, A., Chadwick, M.J., Degris, T., Modayil, J. and Wayne, G., 2018. Vector-based navigation using grid-like representations in artificial agents. Nature, 557(7705), pp.429-433.

[3] - Whittington, J.C., Muller, T.H., Mark, S., Chen, G., Barry, C., Burgess, N. and Behrens, T.E., 2020. The Tolman-Eichenbaum machine: unifying space and relational memory through generalization in the hippocampal formation. Cell, 183(5), pp.1249-1263.

---

### Author Response · Authors · 2023-02-21
**Terminology changes**

We thank the reviewers for their insightful feedback and suggestions. We have decided to use the following terminologies in the revised version of our manuscript.

a) “inference module” instead of “reasoning module”

b) “selection” instead of “attention”

c) “out-of-distribution generalization” instead of “extrapolation”. We would like to note that extrapolation is an extreme form of out-of-distribution generalization. It is true that in high dimensions virtually all generalization can be considered extrapolation, but a more naturalistic interpretation of extrapolation is usually falling “outside the range” of the training data, for which we can consider the boundary as a hypersphere, the center of which is the centroid of the training data, and the radius of which is the maximum of the norms of the vectors from the centroid to each training data point. The training and test data used in our tasks satisfy this condition and the test data is definitely out-of-distribution.

---

### Decision · Action_Editors · 2023-04-11

**Recommendation:** Reject

**Comment:**

### Summary

Deep neural networks are not at generalizing to out-of-distribution data whereas humans can do this easily. This paper proposes a two-part algorithm that aims to improve the OOD generalization performance of neural networks by using grid-like representations and determinantal point processes.

### Decision

The reviewers unanimously agreed to reject this paper. The reviewers have done a good job at addressing the concerns raised by the reviewers. However, in its current form this paper is still not ready for publication. There are two main outstanding concerns raised by the reviewers at the end of the rebuttal period:

1. *Claims are not supported by the experimental results.* This concern is raised by all the reviewers, in particular, the fact that the experiments focus on synthetic tasks, reviewers would like to see the claims to be verified on externally defined benchmark tasks.
2. *Lack of clarity on applicability of the proposed method to higher dimensions.* Raised by reviewers  6GAB and N22L.

In the light of the reviews, I recommend the authors to consider resubmitting the paper after a major update on the paper.


**Audience:**

Yes, the OOD generalization and robustness are important  problems in machine learning. Thus, this paper is relevant for the interests of TMLR community.

**Claims And Evidence:**

The claims made in this paper are only supported with limited synthetic settings. It is not clear whether the proposed selection mechanism over grid codes also aids generalisation in higher dimensions and when more complicated transformations are employed. As a result, the applicability of the proposed approach's to the higher dimensions is unclear. This work does not provide sufficient evidence to support the claim that grid codes are used in the brain or in machine learning.